# TrkB agonist antibody ameliorates fertility deficits in aged and cyclophosphamide-induced premature ovarian failure model mice

Xunsi Qin [1,2], Yue Zhao [3,4], Tianyi Zhang[1,2], Chenghong Yin[5], Jie Qiao [3,4], Wei Guo [1,2 ✉] & Bai Lu [1,2 ✉]

Premature ovarian failure (POF) is a leading cause of women's infertility without effective treatment. Here we show that intravenous injection of Ab4B19, an agonistic antibody for the BDNF receptor TrkB, penetrates into ovarian follicles, activates TrkB signaling, and promotes ovary development. In both natural aging and cyclophosphamide-induced POF models, treatment with Ab4B19 completely reverses the reduction of pre-antral and antral follicles, and normalizes gonadal hormone. Ab4B19 also attenuates gonadotoxicity and inhibits apoptosis in cyclophosphamide-induced POF ovaries. Further, treatment with Ab4B19, but not BDNF, restores the number and quality of oocytes and enhances fertility. In human, BDNF levels are high in granulosa cells and TrkB levels increase in oocytes as they mature. Moreover, BDNF expression is down-regulated in follicles of aged women, and Ab4B19 activates TrkB signaling in human ovary tissue ex vivo. These results identify TrkB as a potential target for POF with differentiated mechanisms, and confirms superiority of TrkB activating antibody over BDNF as therapeutic agents.

[1] School of Pharmaceutical Sciences, IDG/McGovern Institute for Brain Research, Joint Graduate Program of Peking-Tsinghua-NIBS, Tsinghua University, Beijing 100084, China. [2] Beijing Tiantan Hospital, Advanced Innovation Center for Human Brain Protection, Capital Medical University, Beijing 100070, China. [3] Center for Reproductive Medicine, Department of Obstetrics and Gynecology, Peking University Third Hospital, Beijing 100191, China. [4] Research Units of Comprehensive Diagnosis and Treatment of Oocyte Maturation Arrest, Chinese Academy of Medical Sciences, Beijing 100191, China. [5] Department of Internal Medicine, Beijing Obstetrics and Gynecology Hospital, Capital Medical University, Beijing 100026, China. ✉email: wguo@tsinghua.edu.cn; bai_lu@tsinghua.edu.cn

Premature ovarian failure (POF), also more commonly called primary or premature ovarian insufficiency (POI) recently, is a leading infertility disease in humans, affecting 1–5% of women under age 40[1–3]. Women with POF suffer from infertility, a series of menopausal symptoms, as well as some other severe complications such as psychological distress, osteoporosis, auto-immune disorders, ischemic heart disease, and even increased risk of mortality[1–5]. Currently, there is no effective treatment for POF. Clinical care for women with POF involves hormone replacement therapy (HRT) plus psychological support or taking contraceptive drugs for those who are not desiring pregnancy[1,4,5]. In addition to physical and emotional distress, the greatest unmet need for POF women of childbearing age is to bear children who are genetically their own.

The onset of POF has been attributed to premature follicle depletion due to accelerated atresia or altered maturation/recruitment of primordial follicles, leading to lack of folliculogenesis[6], a highly organized and complex process in which small primordial follicles continuously mature into large preovulatory follicles. The pathogenesis of POF is poorly understood. The proposed mechanisms underlying POF vary from chromosomal or genetic alterations, infections, metabolic disorders to autoimmune diseases and iatrogenic factors (ovarian surgery, radiotherapy, chemotherapy, etc.)[1,2,7]. While there is no good genetic model that could be used for drug efficacy studies, two phenotypic models are often used: the "natural aging" model and the "chemotherapy-induced" model. Efforts for the treatment of POF have been directed towards holding ovarian reserve[8–10], activating dormant follicle[11,12], and alleviating follicle loss via potential protective agents (such as melatonin, etc.)[13–16]. Unfortunately, targets and mechanisms for these treatments are largely unknown, and few approaches described above could promote folliculogenesis of residual follicles in patients. Thus, these efforts are not very effective. One interesting study demonstrated that Akt-activating drugs could activate dormant primordial follicles and stimulate follicle growth. However, this treatment involves surgical removal of a piece of the ovary, fragmenting ovary into cubes for culture, followed by Akt stimulator treatment to activate primordial follicles. The activated donor ovarian cubes were then transplanted back to the recipient's ovary, and subsequent retrieval of mature eggs for in vitro fertilization (IVF)[17]. This study is very interesting scientifically, but the procedure is highly complicated and therefore its practical utility may be limited.

Brain-derived neurotrophic factor (BDNF), an extensively studied neurotrophic factor[18,19], is now known as an ovarian endocrine factor as well[20,21]. Substantial evidence suggests that BDNF plays a role in ovarian follicle development[21–23]. In mice, BDNF is expressed and secreted by granulosa cells (GCs) as well as cumulus cells (CCs)[20]. The receptor for BDNF, TrkB, is expressed in ovary tissues in two isoforms: a full-length TrkB (TrkB-FL) with an intracellular tyrosine kinase domain (iTK) for signaling, and a truncated one (TrkB-T1) that lacks iTK and is presumably non-functioning[22]. Ovarian follicles at different stages are illustrated in Supplemental Fig. 7. TrkB-FL is primarily expressed in growing ovarian follicles. In contrast, TrkB-T1 is only expressed in primordial and primary follicles (granulosa cells and oocytes)[22,24,25]. Similarly, BDNF and TrkB have also been detected in human GCs/CCs, and oocytes respectively[21,26–28]. The expression of TrkB-FL and TrkB-T1 in human ovary tissues has not been well documented.

Previous studies indicate that BDNF promotes oocyte maturation of mouse and other animals in vitro[20,26,29,30]. Gene knockout experiments demonstrated that BDNF/neurotrophin-4-TrkB signaling is required for follicle growth and oocyte survival in vivo[31]. The ovaries from both the conventional TrkB-null mice

and newly developed mutant mice lacking all TrkB isoforms exhibited reduced numbers of follicles and abnormal ovaries, reduced GC proliferation, and decreased expression of FSH receptor (FSHR)[24,32,33]. Moreover, oocyte-specific deletion of the Ntrk2 (TrkB) gene revealed early adulthood infertility, with progressive post-pubertal depletion of oocytes accompanied by a loss of follicular organization. All these phenotypes are very similar to those seen in women with POF[25]. In humans, the plasma level of BDNF is decreased in POF patients[34]. Genome-wide association studies (GWAS) analysis have revealed a genetic association between Bdnf (11p14.1) and POF[35]. Treatment with BDNF also promoted meiotic maturation in cultured immature human oocytes[36,37]. Moreover, BDNF stimulated steroidogenesis and increased the proliferation of KGN cells (human granulosa-like cell line) by activating FSHR-mediated signaling[38].

Despite the association between dysregulation of BDNF-TrkB signaling and POF, preclinical and clinical studies suggest that BDNF itself cannot be used as a drug, because of its poor pharmacokinetics[39], limited diffusibility[40], and its activation of another receptor p75[NTR], which often elicits effects different or even opposite to TrkB[38]. Thus, TrkB has never been considered as a drug target for the treatment of POF in previous studies. In all the patents published so far for TrkB agonists or antibodies, POF has never been listed as a disease indication.

Here, we show that a newly developed TrkB agonistic antibody (Ab4B19), with physicochemical properties superior to BDNF[41], can be used to treat POF. Ab4B19, delivered through tail vein injection, successfully engages its target TrkB in the ovary. In two different mouse POF models, Ab4B19 has the capacity to reverse the pathology of POF, rescue ovarian injury, and/or restore the number and quality of oocytes. Single-cell transcriptome analysis suggests that Ab4B19 may elicit similar effects in human cells. Our results support the notion that TrkB may be served as a drug target for POF and demonstrate that the TrkB agonistic antibody Ab4B19 could potentially be useful in treating POF, especially reverse infertility.

## Results

**Ab4B19 activated TrkB signaling in ovarian follicles**. Blood-follicle-barrier (BFB), a molecular sieve with size- and charge-selectivity in ovarian follicles[42], is moderately permeable to mid-sized molecules, such as IgG1 (150 kDa), inter-α-trypsin inhibitor (IαI, 220 kDa), and fibrinogen (340 kDa)[43]. We sought to first examine whether the TrkB agonist antibody Ab4B19 could penetrate into ovarian follicles by crossing BFB and activating TrkB signaling in the ovary. Anti-Müllerian hormone (AMH), a marker of granulosa/cumulus cells around the oocytes, was used to outline the ovarian follicles. Immunostaining was performed using a FITC-tagged anti-rabbit IgG secondary antibody to detect the rabbit monoclonal antibody Ab4B19 in follicles at different time points after its tail vein injection (1 mg/kg; iv) into adult mice. Ab4B19 was consistently present at the follicles at 24 h (Supplementary Fig. 1a), but not at 6 h, after its administration. The immune-reactivity was more abundantly located in granulosa cells (GCs) and oocytes at 48 h (Fig. 1a and Supplementary Fig. 1b). Moreover, we determined the penetration of Ab4B19 across the zona pellucida (Supplementary Fig. 1c). Immunocytochemistry using an anti-rabbit IgG antibody revealed that Ab4B19 not only bound to the surface of oocytes but also were endocytosed into the cytoplasm. Thus, Ab4B19 could penetrate into ovarian follicles in a time-dependent manner. Consistent with the above, the activation of TrkB downstream kinases, Akt1 and ERK1/2, as revealed by anti-pAkt and anti-pERK1/2 antibodies on Western blots, could be seen reliably in the initial 6 h after Ab4B19 administration (Supplementary

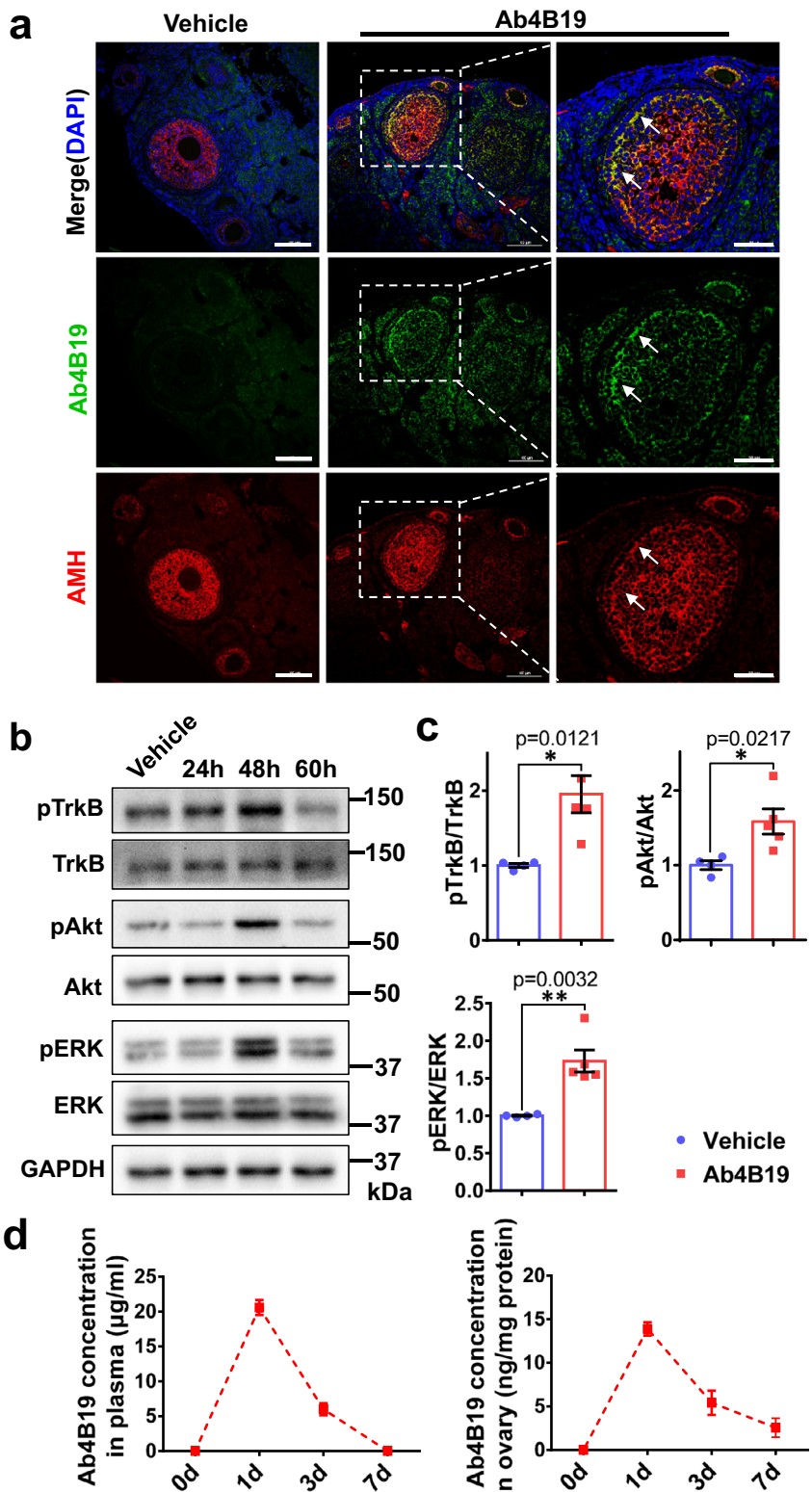

Fig. 1d). With the enrichment of Ab4B19 in ovarian follicles, TrkB signaling was markedly upregulated at 48 h (Supplementary Fig. 1d and Fig. 1b, c). In addition, the pharmacokinetic analysis indicated that the $T_{1/2}$ for Ab4B19 (administered at 1 mg/kg) was approximately 3 days in the blood and ovary tissues (Fig. 1d). These results indicated the engagement of Ab4B19 with its target TrkB in the ovary, paving the way for its potential use for ovarian diseases.

**Ab4B19 promoted oocyte maturation and follicle development.** BDNF promoted oocyte meiosis in culture, as characterized by germinal vesicle (nuclear envelope) breakdown (GVBD), chromosome condensation, and extrusion of the first polar body[20,26]. Our dose-response experiment using cultured oocytes revealed that similar to BDNF (0.2 nM), Ab4B19 (0.2 nM) effectively increased first polar body extrusion, but not GVBD (Fig. 2a, b), and this effect could be blocked by the Trk receptor inhibitor

**Fig. 1 Target engagement of Ab4B19 in ovarian follicles. a** Immunofluorescence staining of ovarian follicles showed that Ab4B19 penetrated across BFB and co-localized with the AMH-positive granulosa cells in ovarian follicles. Ovaries were collected for immunostaining 48 h (48 h) after tail vein injection of Ab4B19 into adult mice. The rabbit monoclonal antibody Ab4B19 was detected with anti-rabbit IgG (FITC, green). AMH was probed with mouse anti-AMH and detected with anti-mouse IgG (TRITC, red). Cell nucleus were labeled with DAPI (blue). White arrows indicate the presence of Ab4B19 in ovarian follicles. Scale bar, 100 and 50 μm (the high magnification). Experiments were repeated three times independently with similar results (four sections/mice each time). **b** Time course of TrkB activation in the ovary, revealed by Western blots, after Ab4B19 administration (1 mg/kg; iv) into 8-week-old mice. Ovary tissues were collected and lysed at different time points after tail vein injection of Ab4B19, and pTrkB, pAkt, and pERK were analyzed. **c** Quantitative plots of TrkB activation and its downstream signaling (pAkt/Akt, pERK/ERK) at 48 h. $N = 4$ mice in the vehicle group and 5 mice in the Ab4B19 group, with the same samples repeated at least twice ($n = 10$). **d** Pharmacokinetics of Ab4B19 in plasma (left) and ovary tissues (right). Ab4B19 was administered into mice (2 months old, 1 mg/kg) by tail vein injection, its concentrations in ovary and plasma at different time points were analyzed by ELISA, and plotted against time ($N = 3$ mice per time point). Data were presented as mean ± SEM. Statistical analyses were carried out by two-tailed student's $t$-test ($*P < 0.05$; $**P < 0.01$; $***P < 0.001$). Vehicle: normal IgG injected. Source data are provided as a Source Data file.

K252a (Fig. 2c). We further determined whether Ab4B19 could promote oocyte ovulation under physiological conditions. After treating proestrus adult mice with Ab4B19 for 3 days, ovary tissues were sectioned and analyzed by H&E staining. Quantitative analysis indicated that there were a lot more corpora lutea in the Ab4B19-treated group, compared with the control (normal IgG treated, 1 mg/kg) group, suggesting that Ab4B19 promoted oocyte maturation (Fig. 2d).

Next, we examined the effect of Ab4B19 on ovarian follicle development using 15-day-old juvenile mice. The ovaries of these pre-pubertal mice contain primarily pre-antral follicles. Thus, the effect of Ab4B19 on pre-antral follicles could be easily determined by counting the follicle numbers. According to the accepted definitions[8], ovarian follicles were categorized as follows: primordial follicles, pre-antral follicles (primary and secondary follicles), and large growing (antral and preovulatory) follicles (See Supplementary Fig. 7 for follicles at various developmental stages). The ovaries were isolated and sectioned for H&E staining 5 days after a single dose of Ab4B19 (1 mg/kg). The representative images and quantification histogram of the different follicle populations are shown in Fig. 2e, f. Ab4B19 significantly facilitated the antral follicles development, but not primordial follicles and pre-antral follicles (Fig. 2f). Meanwhile, Ab4B19 also increased the level of AMH, which was produced by growing follicles ranging from primary to antral follicles (Fig. 2g). In addition, the expression of FSHR, which is required for follicle development and reflective of the biochemical differentiation of growing follicles, was also increased after Ab4B19 treatment (Fig. 2g).

We further determined the phosphorylation of FOXO3a, a key regulator of primordial follicle activation, as well as the expression of DDX4, an oocyte marker mainly expressed in primordial follicles, using 3-day-old ovaries cultured for 2 days. The western blotting analysis did not detect any change in pFOXO3a or DDX4 after Ab4B19 treatment (Supplementary Fig. 2a). Immunofluorescence revealed similar DDX4 staining in control and Ab4B19-treated ovaries (Supplementary Fig. 2b). These results suggest that TrkB activation by Ab4B19 contributed to the oocyte maturation and follicle development but had minimal effect on primordial follicles.

**Ab4B19 attenuated ovarian degradation in NA-POF model.** To determine whether Ab4B19 could be useful for POF therapy, we used the natural aging POF model (NA-POF), which mimics the conditions of progressive ovarian degradation[44]. The schematic timeline of the Ab4B19 treatment is shown in Fig. 3a. We firstly measured whether the concentration of BDNF in the ovary changed as the mice got older. The level of BDNF protein, as determined by BDNF-ELISA, declined steadily from 2 months to 14 months, reaching 0.40 pg/mg protein in 10 months, about half of the level in 2-month-old mice (Fig. 3b). Pharmacodynamic

(PD) analysis for the number of ovulated oocytes per mouse was performed using 12-month-old mice administered with Ab4B19 at various doses (0.1 mg/kg, 0.3 mg/kg, 1 mg/kg). In mice, the average estrous cycle is 4 days and the time from primordial follicle activation to ovulation is about 2–3 weeks. Given that PK ($T_{1/2}$) of Ab4B19 (1 mg/kg) in blood and ovary was around 3 days (Fig. 1d), we decided to administer Ab4B19 through tail vein injection once every 4 days for 16 days. Administration of Ab4B19 to NA-POF (1 mg/kg) elicited a significant increase in ovulation, as measured by the number of ovulated oocytes per mouse (Fig. 3c). Similar to that in the young adult mice, the phosphorylation of Akt1 and Erk1/2 was also increased around 48 h after Ab4B19 administration to NA-POF (Supplementary Fig. 3a, b).

Next, we investigated the mechanisms by which Ab4B19 facilitates ovarian functions. Ab4B19 administration corrected the disrupted estrous cycle, which had been monitored for 12 days until they were caged with males (Fig. 3d), showing an "N" like cycling curve. Ab4B19-treated mice displayed more estrus stage (E), but not metestrus and diestrus stages (M/D) nor proestrus stage (P), as revealed by microscopic analysis of the predominant cell types in the vaginal smears (Fig. 3e). Ab4B19 also improved the morphology and increased the follicle number of the ovary in the NA-POF mouse model (Fig. 3f–h). Quantitative analyses indicated that Ab4B19 also reduced the atretic follicles (Fig. 3h), increased the number of corpora lutea, and restored the development of pre-antral follicles and antral follicles, without affecting primordial follicles (Fig. 3g). Treatment with Ab4B19 also reversed abnormality in blood E2 hormone (Fig. 3i). Interestingly, it did not affect the levels of FSH hormone (Fig. 3j). Finally, the expression of AMH and FSHR was also upregulated after Ab4B19 treatment (Fig. 3k). Taken together, Ab4B19 could halt multiple aspects of ovarian degradation in NA-POF mice.

**Ab4B19 rescued infertility in the NA-POF model.** In the next series of experiments, we determined whether Ab4B19 could rescue infertility in NA-POF mice. The quantity and quality of oocytes were considered critical indicators of a fertility assessment. As shown in Fig. 3c, administration of Ab4B19 (1 mg/kg, once every 4 days) for 16 days increased the number of ovulated oocytes. The same treatment also reduced the proportion of abnormal oocytes in NA-POF mice (Fig. 4a, b). Next, the Ab4B19-treated female mice mated with untreated fertile males were divided into two groups. The first group was euthanized to examine the implantation sites for pregnant estimation around 8 days after vaginal plug detection. Quantitative analysis indicated that NA-POF mice treated with Ab4B19 harbored more embryos, showing higher pregnant capacity compared with untreated NA-POF (Fig. 4c, d). The second group of female mice was kept with male partners for one more week and their fertility rate and the number of pups born were quantified. Health assessment

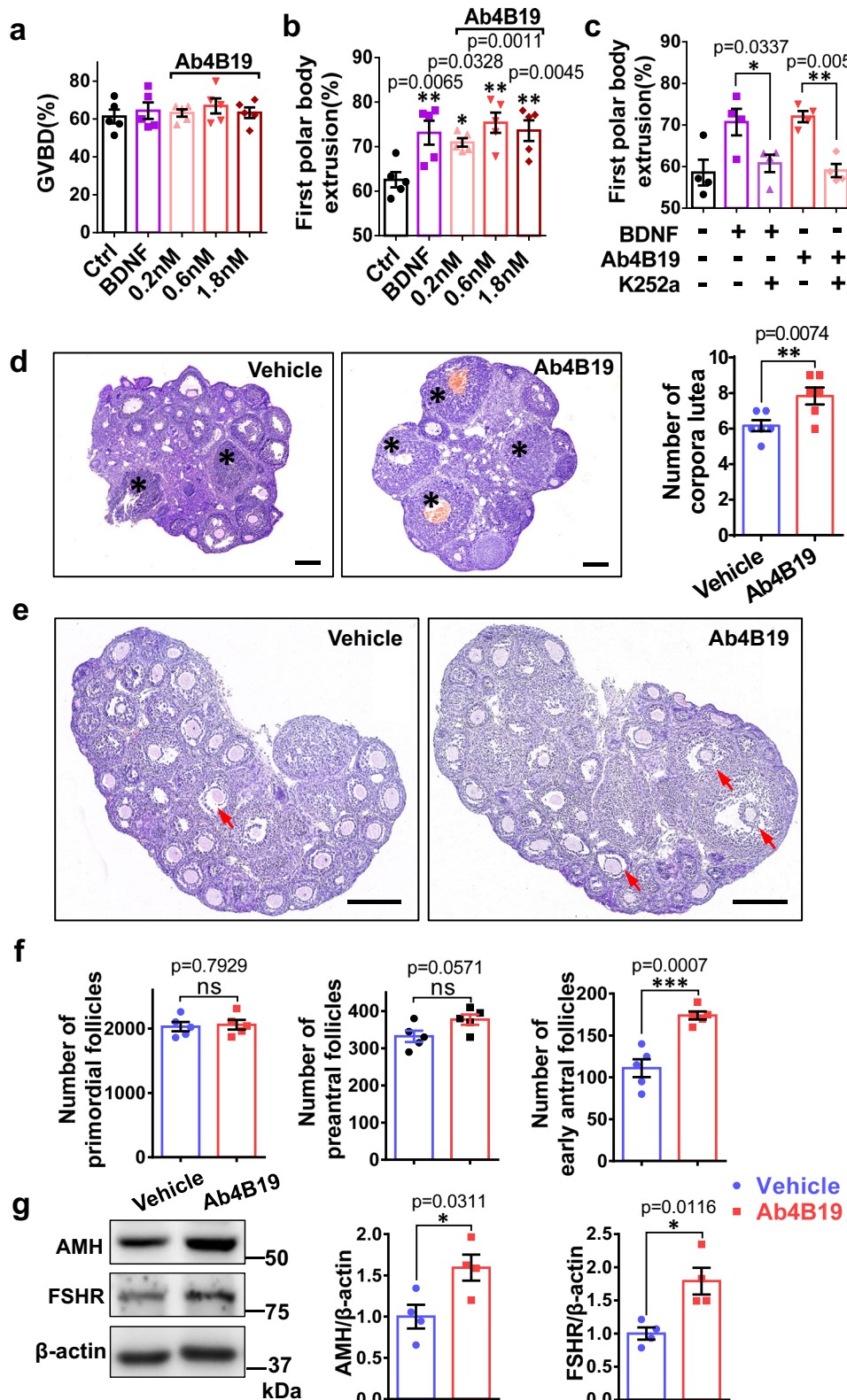

including body weight and appearance was done for these newborn mouse pups. The fertility index, namely the ratio of females delivered offspring/total females, was up to 38.7% for Ab4B19-treated group, and 14.2% for the vehicle-treated group. The average number of newborn pups per mouse reached 3.6 (ranged from 0 to 6) for the Ab4B19-treated group, but only 1.1 (ranged from 0 to 3) in the vehicle-treated group (Fig. 4e). The newborn

pups (age 2-day) derived from Ab4B19-treated mothers had normal body weight (Fig. 4f), and exhibited no malformation, compared with control pups derived from vehicle-treated mothers. Interestingly, the analyses of embryo implantation and litter size indicate that the number of actual birth was generally lower than the number of embryos implanted in the uterus, due most likely to intrauterine abortion or absorption of embryos.

**Fig. 2 Oocyte maturation and pre-antral follicle development stimulated by Ab4B19. a–c** Effect of Ab4B19 on cultured oocytes. Isolated oocytes from 6-8-week-old female mice were cultured with 0.2 nM BDNF, or 0.2, 0.6, and 1.8 nM Ab4B19 with or without K252a Normal mice of the same age were used as the control. GVBD of oocytes (**a** 868 oocytes were used in total) and first polar body extrusion (**b** 826 oocytes were used in total and **c** 652 oocytes were used in total) were measured. ($N = 5$ (**a**, **b**) or 4 (**c**) mice in each condition). **d** Effect of Ab4B19 on the number of corpora lutea in the ovary. Mice of 7-8-week-old were sacrificed and sectioned for H&E staining 3 days after administration of Ab4B19 (1 mg/kg) or normal IgG (1 mg/kg) through tail vein injection. Representative images (left) and quantification (right) of the corpora lutea were presented. Scale bars, 200 μm. The corpora lutea were indicated by asterisks. $N = 6$ mice. **e** Effect of Ab4B19 on the number of antral follicles in developing ovaries. The representative images of H&E staining were captured from the midline histological sections isolated from P15 ovaries 5 days after a single dose of Ab4B19 or normal IgG (1 mg/kg). The red arrows indicated the antral ovarian follicles. Scale bars, 200 μm. **f** Quantification of the effect of Ab4B19 on different types of ovarian follicles for the experiment of (**e**). The number of primordial, pre-antral, or antral follicles was counted and plotted ($N = 5$ mice). **g** Effect of Ab4B19 on the expression of AMH and FSHR. Protein samples from P20 mice administered with Ab4B19 or normal IgG (1 mg/kg) for 5 days were analyzed with Western blot and quantified (right) with β-actin used as a loading control ($N = 4$ mice in each condition). Experiments were repeated two to three times with similar results. Data were presented as mean ± SEM. Statistical analyses were carried out using one-way ANOVA followed by the Dunnett's multiple comparisons test for (**a–c**), or two-tailed student's $t$-test (**d**, **f**, **g**) respectively (*$P < 0.05$; **$P < 0.01$; ***$P < 0.001$; ****$P < 0.0001$; ns not significant). Source data are provided as a Source Data file.

These results together demonstrated the beneficial effects of Ab4B19 on the fertility in the NA-POF model.

Next, a series of safety evaluation experiments were performed in not only the first generation (those derived from Ab4B19-treated mothers) but also the second generation (those derived from males or females whose mothers were treated with Ab4B19) of offspring mice. Supplementary Table 1 summarizes the fertility parameters, which include fertility index, estrous cycle, number of pups born, and mean body weight, for the first-generation mice at 7–8 weeks of age. These pups were completely normal in all these parameters. Supplementary Table 2 shows blood test results, which include cholesterol, glucose, FSH, LH, as well as the morphology of testis, ovary, liver, etc., of the first-generation mice, and the mice from control mothers, at the same age. No difference was found between the two groups. Supplementary Table 3 and Supplementary Table 4 summarize the fertility parameters and blood test results for the second-generation mice at 7–8 weeks of age. Again, the offspring derived from Ab4B19-treated grandmothers were almost the same as those derived from control grandmothers. Finally, we assessed whether the Ab4B19 treatment affected the cognitive functions of the offspring. In the novel object recognition test, there was no difference between the first generation of offspring and those derived from control mice, in either exploration time or discrimination ratio, during both acquisition and test periods (Fig. 4g). Safety evaluation of Ab4B19 for the mothers was performed by examining a number of key blood markers and liver morphology (Supplementary Table 5). Six items were examined and none of them were changed in female mice treated with Ab4B19. Moreover, our previous work also demonstrated that repeated dosing of a TrkB agonistic antibody for 9 months had no obvious off-target effect. There was a small reduction of body weight, as well as an increase in lifespan, which are the expected benefits of activation of the BDNF-TrkB pathway[45].

**Ab4B19 alleviated ovarian injuries in the Cy-POF model.** Premature ovarian failure followed by infertility is one of the most severe side effects of chemotherapies in young female cancer patients, but there is no effective treatment for the gonadotoxic damages in their ovary. Cyclophosphamide (Cy) is a commonly used chemotherapeutic that could induce severe ovary damage and is a recognized risk for POF[46]. Previous reports showed that the depletion of primordial follicles induced by Cy treatment could last for more than 20 days[47,48]. We also found that the number of primordial follicles in Cy-treated animals at 13 days was significantly reduced than that at 7 days after Cy induction in mice (Supplementary Fig. 4a). Therefore, a Cy-induced mouse model (Cy-POF) was used to assess the therapeutic efficacy of Ab4B19 on preserving primordial follicles 7 days after Cy

induction in this study. Figure 5a schematizes the protocol and timeline for generating Cy-POF mice and Ab4B19 treatment. A single dose of Cy (75 mg/kg, 200–300 μl) was administered intraperitoneally to young female mice (6–8-week-old, C57BL/6). After a 7-day induction, the mice were treated with Ab4B19 or vehicle for 6 days (1 mg/kg, iv, once every 3 days). Note that the dosing interval was shortened to 3 days because more drug exposure will benefit cell survival in this acute injury model. H&E staining (midline sections) of ovaries revealed that Ab4B19 treatment significantly alleviated the Cy-induced damage to ovarian follicles, resulting in their normal development (Fig. 5b, c). Quantitative analysis indicated that Ab4B19 treatment attenuated the Cy-induced reduction in the number of primordial follicles, as well as those of the early growing follicles, antral follicles (Fig. 5c). Ab4B19 also reversed the Cy-induced increase in atretic follicles (Fig. 5c). Cy-treated ovaries also exhibited a reduced ratio of corpora lutea, representing less successful ovulation, and this was also rescued by Ab4B19 treatment (Fig. 5c).

While mechanisms underlying chemotherapy-induced ovarian damage are not fully understood, its direct toxicity to growing follicles by apoptosis is apparent[49]. Western blotting revealed that Ab4B19 treatment increased the ratio of BCL2 to BAX protein level (Supplementary Fig. 4b) and reduced the expression of cleaved-caspase-3 (Fig. 5d), suggesting inhibition of Cy-induced apoptosis. Immunostaining of cleaved-caspase-3 identifies apoptotic follicles. The fluorescence signals were dramatically increased in the ovarian sections derived from Cy-POF mice, but not in those from POF mice treated with Ab4B19 (Fig. 5e). Quantification of follicles in different developmental stages indicated that the apoptotic index (caspase-3-positive follicles/total follicles) in pre-antral and early antral stages were rescued in Ab4B19-treated ovaries, compared with the vehicle-treated ovaries (Fig. 5e, f).

**Ab4B19 reversed infertility in the Cy-POF model.** We next examined the effect of Ab4B19 on fertility in Cy-POF mice. Because of Cy-induced gonadotoxic damage to the ovary, the expression of AMH was decreased in Cy-treated ovary compared with control ovary, but such a decrease was reversed after Ab4B19 treatment (Fig. 6a, b and Supplementary Fig. 4c). Cy-induced injury in ovaries was accompanied by an increase in FSH but a decrease in E2 hormone in the serum. Remarkably, Ab4B19 treatment reversed the levels of these two hormones close to normal levels (Fig. 6c, d). Moreover, the Cy-POF mice exhibited a reduced number of ovulated oocytes and an increased proportion of abnormal oocytes, and treatment with Ab4B19 reversed these changes as well (Fig. 6e–g). By comparison, with the same dose but more dosing times than Ab4B19, treatment with BDNF failed to elicit an increase in the number of ovulated oocytes or decrease

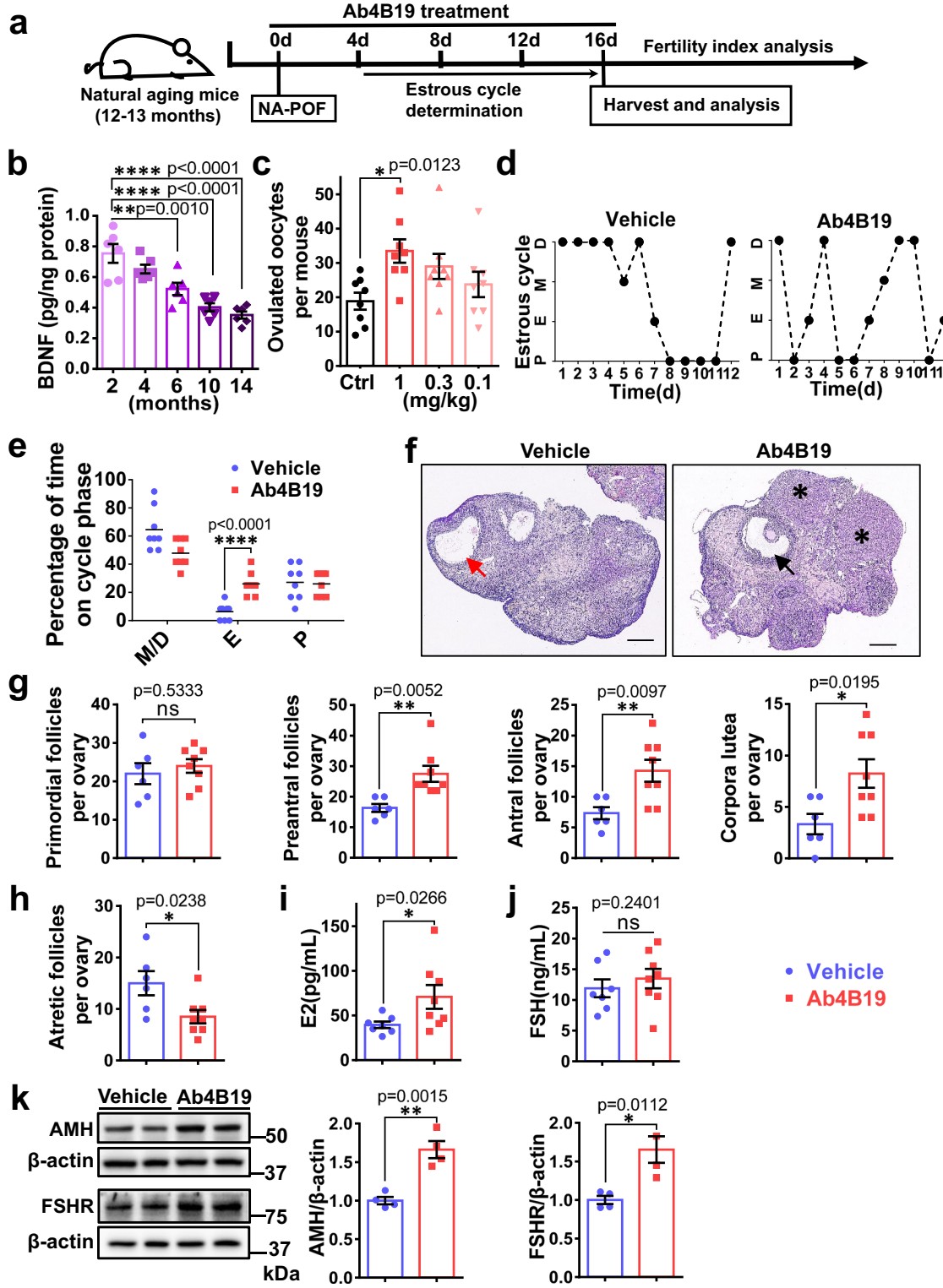

in the proportion of abnormal oocytes (Supplementary Fig. 5), demonstrating the superiority of Ab4B19 over BDNF as a therapeutic agent for Cy-POF.

To further determine whether Ab4B19 improved the fertility of Cy-POF mice, mating trials were performed. It has been shown that treatment with 75 mg/kg Cy led to a short-term decrease in fertility because of a decreased number of growing follicles, but the fertility returns to normal due to a continuous supply of the remaining primordial follicles[50]. In the longer

term, the fertility of mice would decline again due to the gradual exhaustion of primordial follicles[15]. It is therefore important to choose the right time window to reveal the fertility changes in the Cy-POF model. We mated Ab4B19-treated female mice with untreated fertile males 3 days after the final dosing of Ab4B19 (1 mg/kg, once every 3 days for 6 days, Fig. 5a), a time window in which Cy should elicit it's a clear decrease in fertility (i.e., 7 days after treatment with Cy). We found that all female mice were successfully mated within

**Fig. 3 Therapeutic effects of Ab4B19 on ovarian degradation, acyclicity, and hormone balance in NA-POF model. a** Experimental design of Ab4B19 treatment and analysis in NA-POF model. **b** BDNF concentration in ovaries at different ages, from 2 months to 14 months. ($N = 6$ mice per group). **c** Pharmacodynamic analysis showing the number of ovulated oocytes per mouse treated with increasing doses of Ab4B19 from 0.1 mg/kg to 1 mg/kg. ($N = 8$ mice per group). **d–j** Aged mice of 12–13 months old were treated with Ab4b19 or IgG (1 mg/kg) every 4 days through tail vein injection. After 16-day (four doses) treatment, ovaries and serum were collected for analysis. **d** Representative estrous cycles of NA-POF mice treated with Ab4B19 (left) or normal IgG (right). P proestrus stage, E estrus stage, M metestrus stage, D diestrus stage. **e** Effect of Ab4B19 on time spent in various estrous stages. Stages of the estrous cycle were determined by microscopic analysis of the predominant cell types in the vaginal smears. ($N = 8$ mice per group). **f** Representative images of H&E staining of ovaries from NA-POF mice treated with Ab4B19 or vehicle. Red arrow: atretic ovarian follicle; Black arrow: antral ovarian follicle; Asterisk: corpora lutea. Scale bars, 200 μm. **g, h** Quantitative analysis of the primordial follicles, pre-antral follicles, antral follicles, and corpora lutea, as well as the atretic follicles. ($N = 6$ ovaries in vehicle group and 8 ovaries in Ab4B19 group). The serum levels of estrogen (**i**) and FSH (**j**) were measured by ELISA ($N = 7$ mice in vehicle group and 8 mice in Ab4B19 group). **k** Representative Western blot and quantitative analyses of FSHR and AMH expression. β-actin was used as a loading control. $N = 4$ mice in each condition, with the same samples repeated at least twice ($n = 8$). All experiments were repeated at least three times. The data were shown as the mean ± SEM with individual values. Statistical analyses were carried out using one-way ANOVA followed by the Dunnett's multiple comparisons test for (**b, c**), or two-tailed student's t-test (**e, g–k**), respectively (*$P < 0.05$; **$P < 0.01$; ***$P < 0.001$; ****$P < 0.0001$; ns not significant). Source data are provided as a Source Data file.

1 week after the male mice were introduced and got pregnant (Supplementary Table 6). However, the Cy-POF mice exhibited a reduced litter size (numbers of pups per litter), compared with the control mice. Importantly, this number was reversed by Ab4B19 treatment (Fig. 6h). To further examine the long-term protective effect of Ab4B19 on the primordial follicle pool, these female mice were mated a second round ~5 weeks after the first-round parturition. Treatment with Ab4B19 again resulted in a significant increase in fertility in the Cy-treated female mice when they were mated in the second round (Fig. 6i). On average, there were ~10 babies/mother for control, 6 babies/mother for Cy, and 9 babies/mother for Cy-Ab4B19 groups in both first and second rounds, indicating clearly that the TrkB antibody treatment improved fertility. Moreover, the six female mice in the Ctrl group had given altogether 21 times of birth, averaging 3.5 births/mouse before termination (in 9 months). In Cy-POF females, the Vehicle and Ab4B19-treated groups actually had completed 15 times of birth ($15/6 = 2.5$ births/mouse) and 18 times of birth ($18/6 = 3$ births/mouse) before termination, respectively. Thus, Ab4B19 not only increased the number of pups/birth but also the number of births in their adult lives (9 months). These results suggest that the gonadotoxic damage of growing follicles in the Cy-POF model, and both the short-term and long-term defects of fertility could be well rescued by Ab4B19 treatment.

**Effects of Ab4B19 on human ovarian cells.** BDNF has been shown to modulate granulosa cell proliferation and steroidogenesis in mice by activating the FSHR/cAMP/PKA/CREB signaling. It is imperative that the utility of Ab4B19 be examined in human ovarian cells. Due to the difficulties in obtaining ovarian tissues from aged women or POF patients, the human granulosa cell line (KGN cells) was employed. KGN cells, established by long-term culture of human ovarian granulosa cell carcinoma, have steroidogenic activities and express functional FSH receptors. Thus, this cell line is considered a very useful model to explore the physiological regulation of human granulosa cells[51,52]. We found that Ab4B19 at a concentration as low as 0.2 nM promoted the survival of KGN cells, as measured by Cell Counting Kit-8 (CCK-8) (Fig. 7a). Application of Ab4B19 to KGN cells in culture also stimulated CREB phosphorylation after 24-h treatment (Fig. 7b). These results provide preliminary evidence for the efficacy of Ab4B19 on human ovarian cells.

To further explore the clinical relevance of TrkB activation in POF, we reanalyzed our single-cell RNA-seq experiments for BDNF and TrkB mRNA expression in human ovarian follicles. Data from human oocytes and corresponding granulosa cells

(GCs) spanning five follicular stages were obtained. The dynamic expressions of BDNF and TrkB transcripts were examined using principal component analysis (PCA) (Supplementary Table 7). We found that BDNF transcripts were expressed in GCs of all five follicular stages, with an increased expression level in oocytes from secondary to antral follicles (Fig. 7c). More importantly, the expression pattern of the two TrkB isoforms in human oocytes and GCs suggests a similar functional mechanism in the ovary. TrkB-T1 mRNA expression exhibited a gradual decline during development in GCs (Fig. 7d, left) and oocytes (Fig. 7e, left). In contrast, the expression of TrkB-FL mRNA experienced a transient upregulation in GCs of primary follicles (Fig. 7d, right) but a sustained increase in oocytes from the stage of secondary follicles (Fig. 7e, right). Thus, the changes in the expression of TrkB-T1 and TrkB-FL in human oocytes are similar to those in mice.

To determine the expression of BDNF protein and the impact of aging, we performed BDNF immunostaining with ovarian sections derived from young and aged females. In clinics, POF patients seldom undergo surgery. Here the young (ages 29–35) and aged (61–64) female groups were used as the high- and low-fertility group respectively. It is generally believed that the follicle number is dramatically reduced after menopause. However, there are occasional reports that at age 60 or even 70, women could still get pregnant, suggesting that some of their follicles might still be functional after amenorrhea. With careful examination, we could identify a few small follicles (with diameter less than 100 μm) with clear structural characteristics of primordial or primary follicles and co-labeling with an established marker DDX4/MVH (Fig. 7f). We found that BDNF expression in follicles from aged women is significantly lower than that in follicles from young ones (Fig.7f, g and Supplementary Fig. 6b). The levels of BDNF protein from individual samples are shown in Supplementary Fig. 6a. To further investigate whether Ab4B19 could increase TrkB activation in human tissues, we performed an ex vivo experiment. Human ovarian tissues with visible follicles were dissected from female donors who underwent surgery for endometrial stromal sarcoma, fragmented into small cubes, and incubated in 48-well tissue culture plates with culture medium at 37 °C. Then they were treated with 1 nM BDNF, 3 nM normal IgG, or 3 nM Ab4B19 for 35 min and lysed for Western blotting analysis. Similar to BDNF, treatment with Ab4B19 elicited a marked increase in phosphorylated Erk, a sensitive downstream signal of TrkB activation (Supplementary Fig. 6c, d). These results reveal the translational potential of the present study: Ab4B19 may have similar therapeutic effects in humans as those in the mouse models.

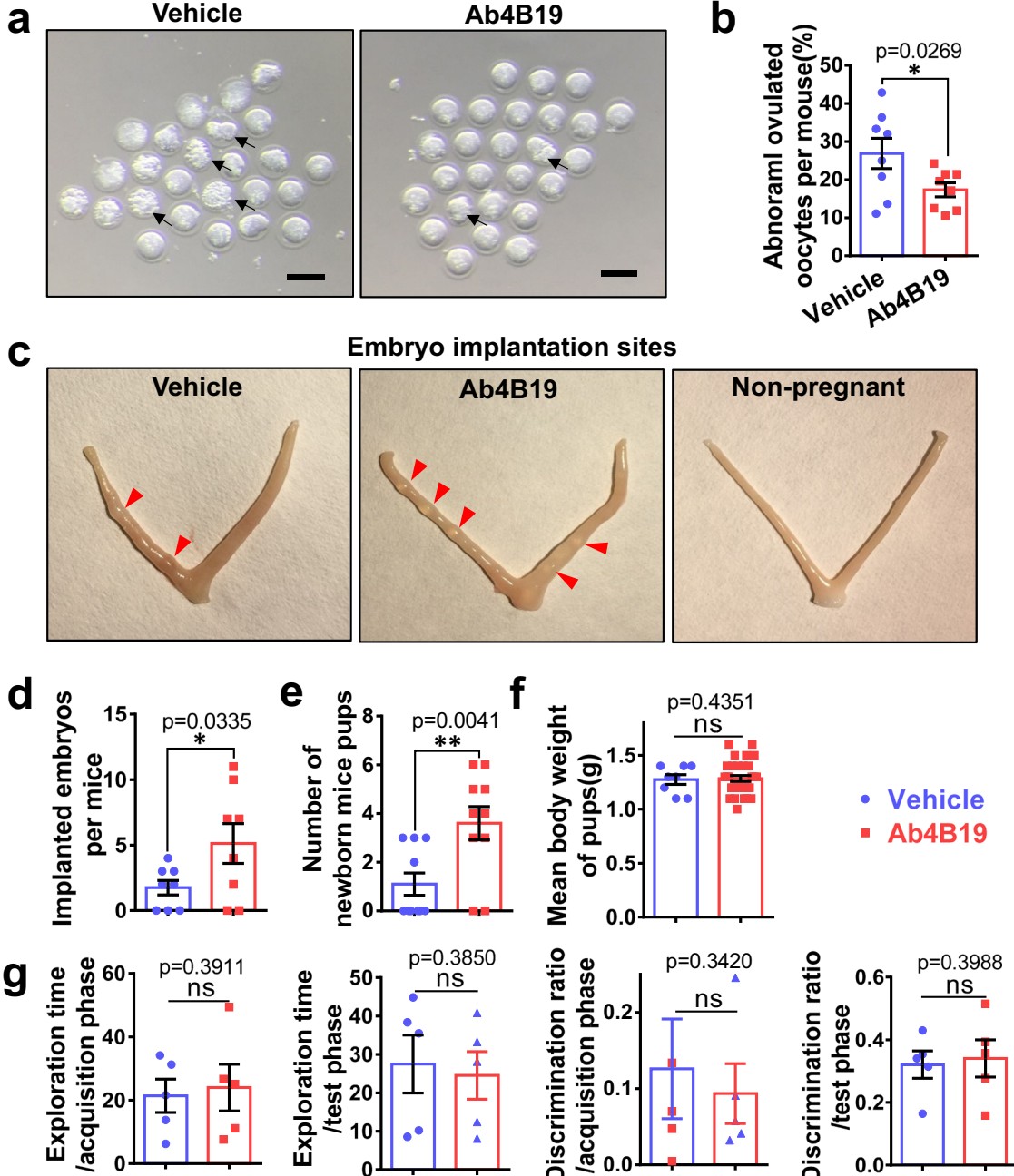

**Fig. 4 Effect of Ab4B19 on fertility in NA-POF mice. a** Representative photomicrographs of oocytes from NA-POF mice treated with or without Ab4B19. Note that there are much more abnormal oocytes in a vehicle-treated group than the Ab4B19-treated group. Black arrow: abnormal oocytes. Scale bar, 100 μm. **b** Quantification of abnormal oocytes by the percentage of abnormal oocytes/total ovulated oocytes per mouse. $N = 8$ mice per group. **c** Representative images of the embryo implantation sites from NA-POF mice in vehicle-treated group, Ab4B19-treated group, and non-pregnant mice used as the control. These mice were sacrificed 7.5–8.5 days after vaginal plug detection. The red arrowheads indicated the implanted embryos in the uterus. **d** Quantification of the number of implanting embryos per mouse. $N = 8$ mice per group. **e, f** Quantitative analysis of fertility parameters. Female mice were treated with vehicle or Ab4B19 for a period of over 2 months until they got pregnant and their pups were born. Total numbers (**e**) and body weight (**f**) of newborn pups in vehicle and Ab4B19-treated groups ($N = 10$ female mice in **e**; $N = 8$ pups in vehicle group and $n = 31$ pups in Ab4B19-treated group). **g** Novel object recognition test for the first-generation offspring from Ab4B19-treated NA-POF group. Normal mice of the same age were used as the control. $N = 5$ mice per group. Data were presented as mean ± SEM. Statistical analyses were carried out by two-tailed student's $t$-test (*$P < 0.05$; **$P < 0.01$; ***$P < 0.001$; ns not significant; **b**, **d**–**g**). Source data are provided as a Source Data file.

## Discussion

As of today, there is no FDA-proved drug for POF, despite huge unmet needs[1,2,16]. In this study, we have provided evidence that an agonistic antibody (Ab4B19) targeting the BDNF receptor TrkB in ovarian follicles could be a promising strategy for POF treatment. Ab4B19, which is superior to BDNF in diffusibility, PK and receptor specificity, activated TrkB and its downstream signaling in ovary, and promoted oocyte maturation and ovarian follicular development. In the natural aging-induced mouse model (NA-POF), treatment with Ab4B19 rescued ovarian degradation and infertility. In the cyclophosphamide (Cy) induced POF model (Cy-POF), Ab4B19 repaired ovarian damage,

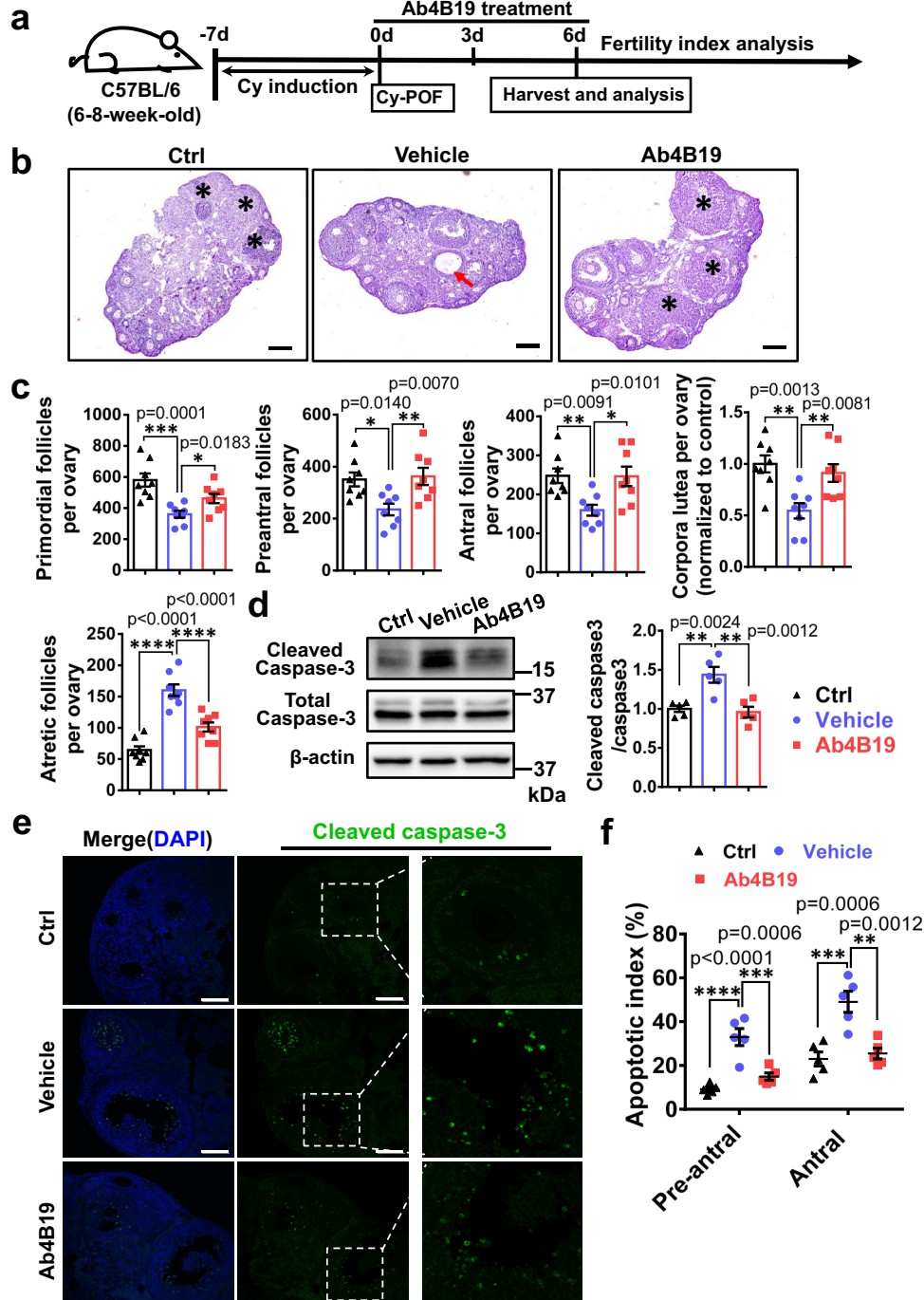

**Fig. 5 Alleviation of ovarian defect and gonadotoxicity by Ab4B19 in Cy-POF model. a** Experimental design of Ab4B19 treatment and analysis in Cy-POF mice. **b** H&E staining ovaries (midline sections) harvested from a normal female mouse (left) or a Cy-POF mouse treated with vehicle (center) or Ab4B19 (right) for 6 days. Asterisks: the corpora lutea; Red arrow: atretic ovarian follicle. Scale bars, 200 μm. **c** Quantification of primordial, pre-antral, antral follicles, atretic follicles, and corpora lutea per ovary in normal mice (Ctrl), and Cy-POF mice treated with vehicle or Ab4B19. $N = 8$ ovaries per each condition. Unless stated otherwise, statistical analyses for comparison among three or more groups in this and all the following figures were carried out using one-way ANOVA, followed by Dunnett's multiple comparisons tests. **d** Western blotting showing the expression of cleaved and total caspase-3. The apoptotic level was determined by the normalized ratio of the cleaved to total caspase-3. $N = 5$ mice per group. **e** Immunofluorescence images of ovaries stained with cleaved caspase-3 antibody (Green: cleaved caspase-3, Blue: DAPI). Images in the red frames in the center row are magnified and shown on the right. Scale bars, 200 μm. **f** Quantification of apoptosis in pre-antral and antral follicles. Apoptosis index was expressed as apoptotic/total follicles ($N = 5$ mice per group). Data were all presented as mean ± SEM. Statistical analyses were carried out using one-way ANOVA followed by the Dunnett's multiple comparisons test (**c**, **d**, **f**) (*$P < 0.05$; **$P < 0.01$; ***$P < 0.001$; ****$P < 0.0001$). Source data are provided as a Source Data file.

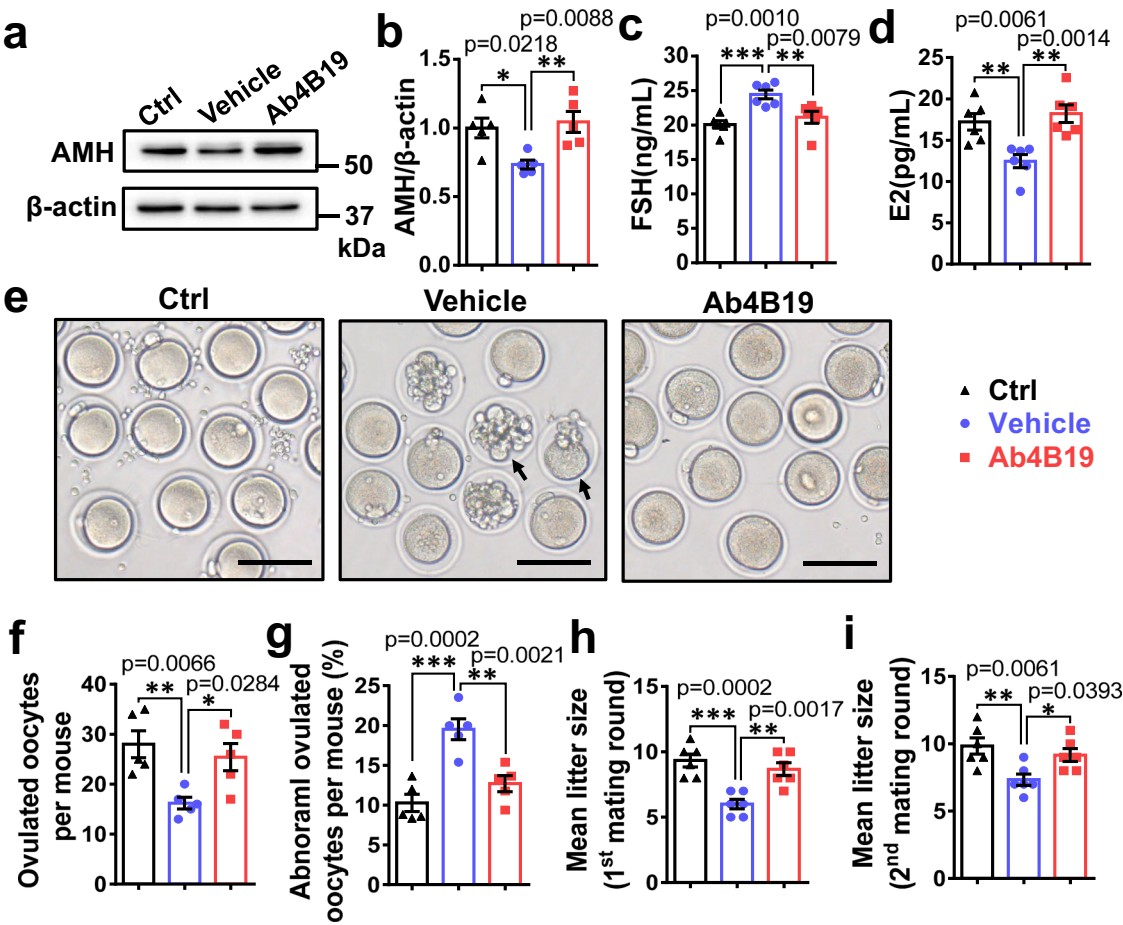

**Fig. 6 Improvement in fertility by Ab4B19 in Cy-POF model. a** Regulation of AMH expression by Cyclophosphamide and Ab4B19 in the mouse ovary. Cy-POF mice were treated with Ab4B19 as shown in Fig. 5a. The lysates of ovarian tissues from three groups, control (Ctrl), Cy-POF mice treated with or without Ab4B19 (saline solution, or normal IgG, Ab4B19) were processed for Western blotting to detect the expression of AMH. **b** Quantification of AMH expression levels (ratio to β-actin). The levels of the control group were normalized to 1. $N = 5$ mice per group. Levels of FSH (**c**) and estradiol (E2) (**d**) were measured by ELISA from the sera collected from the three groups. ($N = 6$ mice per group). **e** Representative photomicrographs of oocytes from the three groups. Note that there were more abnormal oocytes in Cy-POF (vehicle) group than the control group, and treatment with Ab4B19 rescued the deficit. Black arrow: abnormal oocytes. Scale bar, 150 μm. **f** Quantification of the ovulated oocytes per mouse in each treatment group. ($N = 5$ mice per group). **g** Abnormal oocytes. The percentage of abnormal oocytes/total ovulated oocytes per mouse was calculated for each treatment group. ($N = 5$ mice per group). **h** Mean litter size in the first round of mating in control, vehicle, or Ab4B19-treated groups within 4 weeks after the administration with Ab4B19 or normal IgG ($N = 6$ mice per group). **i** Mean litter size in the second round of mating, measured for the same female mice ~5 weeks after the first-round parturition ($N = 6$ mice per group). Data were all presented as mean ± SEM. Statistical analyses were carried out using one-way ANOVA followed by the Dunnett's multiple comparisons test (**b**–**d**, **f**–**i**) (*$P < 0.05$; **$P < 0.01$; ***$P < 0.001$). Source data are provided as a Source Data file.

inhibited follicular apoptosis, and restored the number and quality of oocytes. The potential effects of Ab4B19 on ovarian follicles at various developmental stages, as well as those on the NA-POF and Cy-POF models, are summarized in Supplemental Fig. 7. Further, single-cell RNA-seq analysis of human ovarian cells revealed a significant presence of BDNF in GCs and an increased expression of TrkB in oocytes as they mature. Ab4B19 could also induce TrkB signaling and enhance survival in cells derived from human GCs. These results together have demonstrated the beneficial effects of Ab4B19 on animal POF models and provided a foundation for further testing its therapeutic potential in treating human POF.

Given the lack of therapy for POF, current treatment strategies have focused on indirect or symptomatic treatments, such as alleviating follicle loss or restoring hormonal balance[10,16]. These approaches are generally not very effective. Hormone replacement therapy (HRT) could reduce some of the POF-associated complications and improve the quality of life for women with POF[53]. Unfortunately, these treatments are accompanied by an increased risk of cancer[54,55]. Moreover, there is no standard HRT regimen, which should be individualized according to the diagnosis[4,5]. Some protective agents, such as melatonin, sphingosine-1-phosphate (S1P), ceramide-1-phosphate (C1P), etc., appear to exert positive effects on gonadotropin recovery and restore fertility. How these agents achieve their "protective" effects remain a mystery. One limitation of S1P, an inhibitor of ceramide-promoted cell death, is that it must be injected directly into the ovary to avoid disturbing other physiologic mechanisms[56]. Stem cell therapy has been controversial because of safety concerns on stem cell transplantation into women's ovary[10]. Most importantly, none of the treatments highlighted above have a clear mechanism or target for the illness, making it difficult for further development into effective medicines.

The TrkB antibody strategy used in the present study may have several unique advantages, in comparison with the existing treatments. First, expression studies, cell culture work, and in vivo animal work have all supported the rationale for the protection/

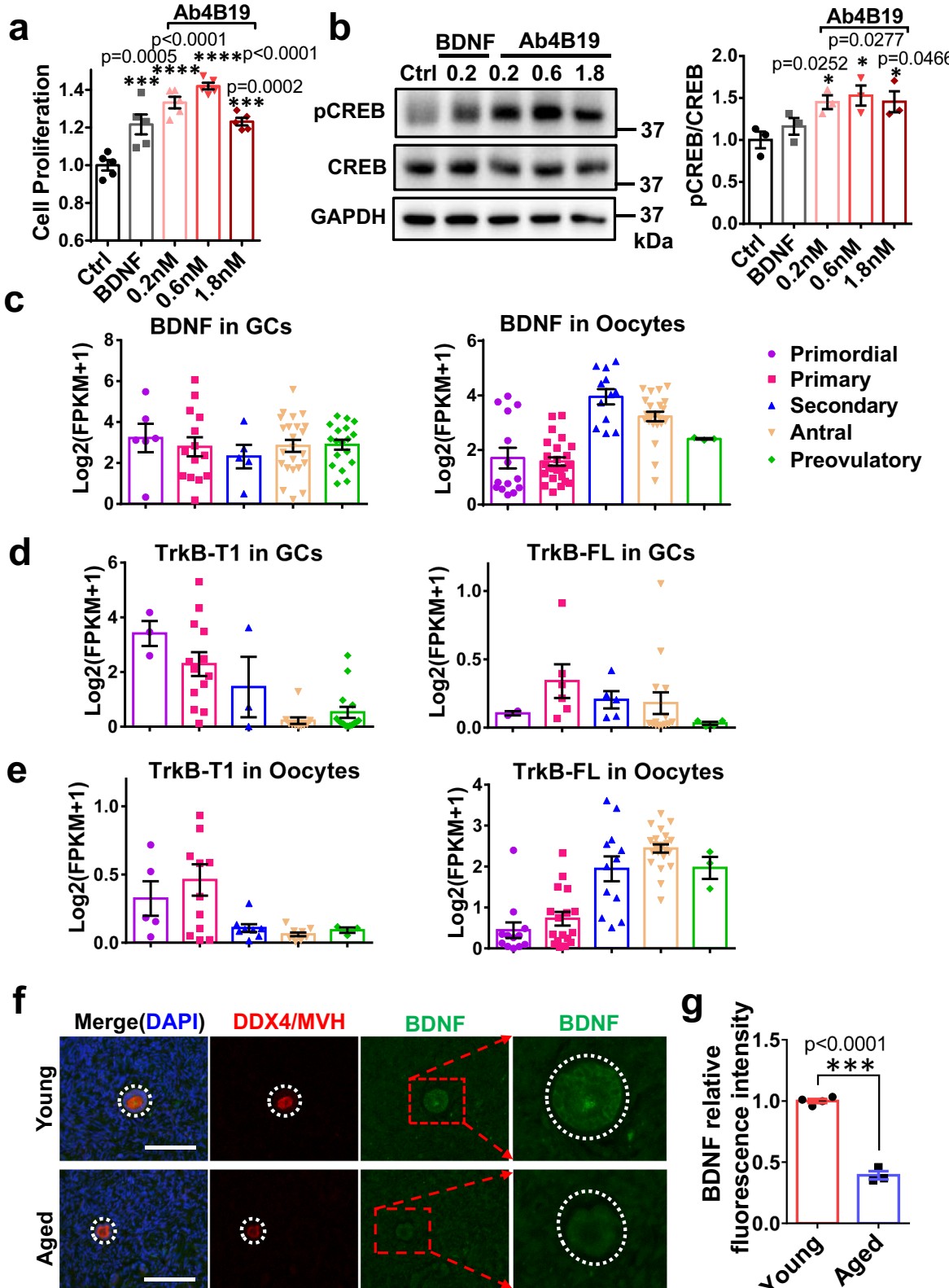

repair mechanism mediated by the BDNF-TrkB signaling pathway. Second, Ab4B19 reverses infertility by preserving folliculogenesis in two animal models of POF. Third, Ab4B19 also exhibits excellent drug-like properties, overcoming the shortcoming of BDNF. Finally, Ab4B19 did not induce any visible side effects in mice. Therefore, Ab4B19 has demonstrated its potential to be a candidate drug.

Once diagnosed, POF patients usually display infertility, a devastating consequence of the disease. These patients have a very low probability of spontaneous pregnancy and do not respond to traditional therapies. However, most of the POF cases are not true "ovary failures": nearly 75% of women with POF have detectable follicles in their ovary. The animal models in our study could not represent all types of human POF/POI, which is a heterogeneous

**Fig. 7 Role of BDNF-TrkB in the human ovary. a** Effect of Ab4B19 on the proliferation of human KGN cells. KGN cells were cultured for 72 h and then treated with Ab4B19 for 24 h. Cell proliferation was measured by the viability analysis on 96-well plates after treatment with BDNF (as a positive control, 0.2 nM) and Ab4B19 (0.2, 0.6, or 1.8 nM) for 48 h ($N = 5$ per group). **b** Activation of CREB by Ab4B19 in KGN cells. Cultured KGN cells were treated with BDNF (0.2 nM) and Ab4B19 (0.2, 0.6, or 1.8 nM) for 24 h, and harvested for pCREB assay. Phosphorylated CREB (Ser133) and total CREB were examined by Western blot. Quantification of pCREB/CREB was plotted on the right. The levels of vehicle treatment were normalized to 1. $N = 3$ repetitions per condition. **c–e** Transcriptional analysis of the expression of BDNF and TrkB in human GCs and oocytes. The raw data were derived from Zhang et al. (Zhang, Yaoyao, et al. Molecular Cell 72.6 (2018): 1021-1034.). *Bdnf* and *TrkB* genes with fragments per kilobase per million (FPKM) >1 in at least one cell were analyzed, and expression levels of each gene were plus one then log2 transformed in the following analysis. **c** Bar plots showing the relative expression levels (log2 [FPKM + 1]) of BDNF mRNA from the primordial to preovulatory ovarian follicle stages. **d** Expressions of TrkB-T1 and TrkB-FL mRNAs in human GCs from five different developmental stages of ovarian follicles. **e** Expressions of TrkB-T1 and TrkB-FL mRNAs in human oocytes from five different developmental stages of ovarian follicles. **f** Typical image of BDNF expression in the human ovary from young and aged people. BDNF-immunoreactivity was detected by a rabbit anti-BDNF antibody followed by anti-rabbit IgG (FITC, green). Co-labeling with a mouse anti-VASA antibody followed by anti-mouse IgG (TRITC, red) was performed to identify follicles. The outlines of follicles were marked using dotted white circles. Scale bar, 100 μm. **g** Relative fluorescence intensity of BDNF expression in young (29–35 years old) and aged (61–64 years old) groups. ($N = 4$ people in the young group and 3 people in the old group). Data were all presented as mean ± SEM. Statistical analyses were carried out using one-way ANOVA followed by the Dunnett's multiple comparisons test (**a**, **b**), or two-tailed student's *t*-test (**g**), respectively (*$P < 0.05$; **$P < 0.01$; ***$P < 0.001$; ****$P < 0.0001$). Source data are provided as a Source Data file.

disorder with 70% idiopathic. Moreover, POF patients who had been through chemotherapy (10–15%) for several years may also be different from Cy-POF mice with acute onset[16]. It would be interesting to determine the effect of Ab4B19 on mice that had been treated with Cy for different lengths of time. The main goal of the present study was to demonstrate the feasibility of our therapeutic strategy: to promote the residual follicular development and maturation with the TrkB agonistic antibody and protect damaged follicles (such as Cy-induced POF).

Our results imply that TrkB antibody can be used to treat different types of POF/POI patients regardless of their etiology or onset time, as long as they have residual follicles. Whether this strategy is truly effective in certain subtypes of POF patients will need to be tested through clinical trials. The NA-POF mice exhibit the ovarian recession with a decrease, but not elimination, in the number of ovarian follicles and a low-fertility rate, mimicking the human condition. Treatment with Ab4B19 rescued ovarian degradation and increased the number of pre-antral and antral follicles in the NA-POF mice, accompanying increases in blood E2 and ovarian expression AMH. The recovery of fertility is the key in evaluating therapeutic efficacy and the best way to determine fertility is to analyze the number of pups delivered by pregnant females. The ratio of the females that delivered offspring to total females was up to 38.7% in the Ab4B19 treatment group, compared with 14.2% in the control group. Finally, the implantation sites of the mouse uterus were also examined to confirm that Ab4B19 increased the pregnancy rate. Taken together, Ab4B19 may help improve the fertility of POF patients with residual follicles by promoting folliculogenesis.

Cyclophosphamide (Cy) causes ovarian dysfunction including accelerated loss of ovarian reserve, folliculogenesis dysregulation, and steroid disorder[8,15]. We found that Ab4B19 increased the number of primordial follicles, pre-antral follicles, and antral follicles in the Cy-POF mice. For the growing follicles and antral follicles, Cy-induced toxicity leads to the apoptosis of ovarian follicles. Although the apoptosis induced by Cy through DNA damage generally occurs within 2 days, the damage of granulosa cells would have a secondary effect on the development time of growing follicles, resulting in atresia of growing follicles and long-lasting apoptosis in follicles. This is why apoptotic signals could still be detected about 2 weeks after Cy treatment. Similar results were reported by others[15]. In brief, our results and previous studies[15,50] showed that 75 mg/kg Cy treatment induced acute (1–2 weeks after treatment) damage (apoptosis, Fig. 5d–f) as well as chronic (more than 3 months) POF-like damage to the ovary (Fig. 6i). Ab4B19 reversed both processes through Akt and Erk-mediated inhibition of apoptosis, thereby maintaining normal ovarian follicle development and significantly improving the fertility of Cy-POF mice.

The protective/repair mechanism for Ab4B19 regulation of primordial follicles might be different from those for growing follicles. A proposed mechanism for Cy-induced loss of ovarian follicle pool is the excessive activation of primordial follicles through the PI3K-Akt pathway, resulting in a "burnout" effect and ovarian follicle depletion[8]. Given that primordial follicles express primarily the "non-functional" TrkB-T1 but not TrkB-FL, it is less likely that Ab4B19 attenuates the PI3K-Akt pathway in the primordial follicles per se. Extensive studies suggest that the AMH level is critical for maintaining the inactive state of primordial follicles[57]. Previous studies have also shown that AMH prevents the over-activation of primordial follicles in Cy-POF mice[9]. In this study, we found that Cy treatment induced a significant reduction in various developing follicles and AMH levels, leading to a continuous loss of primordial follicles. The number of primordial follicles in Cy-treated animals at 13 days was significantly reduced than that at 7 days after Cy induction in mice (Supplementary Fig. 4a). Thus, the consumption of primordial follicles in Cy-treated ovaries would always be higher than that in untreated ovaries, until the AMH level in Cy-POF mice returns to normal, and the duration may be much more than 13 days. Our results that the delayed administration of Ab4B19 (7 days after Cy treatment) partially rescued the ovarian dysfunction in Cy-POF mice indirectly verify that there is secondary Cy-induced damage. Similar experiments in previous reports also showed a long-lasting depletion of primordial follicles induced by Cy treatment: (1) Depletion of primordial follicles induced by 50 mg/kg Cy lasted for at least 20 days in 18-week-old mice[47]; (2) Depletion of primordial follicles caused by 150 mg/kg Cy treatment on day-21 was much more obvious than that on day-7 (~45 vs. 20%)[48]. We, therefore, propose that Ab4B19 may regulate primordial follicles (Fig. 5c) indirectly by maintaining AMH levels, which were elevated in growing follicles in the entire ovary (Fig. 6a, b). In addition to PI3K-Akt mediated "burnout" (direct) and disinhibition by AMH reduction (indirect), several other pathways that induced indirect effects on primordial follicle pools were also summarized by Spears et al. [58]. Further work is necessary to delineate how Ab4B19 regulates primordial follicles.

As a candidate drug for pre-pregnancy women, TrkB agonistic antibody must be carefully evaluated for its safety. During the entire course of Ab4B19 treatment (1 mg/kg, 16 days, once every 4 days), we could not observe any malformation in the animals, except lower body weight after Ab4B19 treatment, a phenomenon

consistently observed in previous studies using TrkB agonistic antibodies[59]. For safety evaluation on embryos, we found no abnormalities in 8-day-old embryos when we examined the implantation sites in the uterus. Moreover, the concentration of Ab4B19 during the period of embryo implantation (about 9 days after its final dosing) was reduced to a very low level (down to about 10%), which may not be sufficient to activate TrkB signaling. Further, health assessments, as well as a detailed examination of the reproductive system of the mice derived from Ab4B19-treated mothers, showed no anomalies. We examined the fertility, as well as some biochemical indices and the gonadal hormones in the blood, in the first- and second-generation offspring mice. There is no visible body malformation and weight change in these mice. Given the important roles of BDNF-TrkB signaling in the brain, we examined the effect of Ab4B19 on novel object recognition, a cognitive function known to be regulated by BDNF[60], in the first-generation offspring mice. Again no anomaly was found. A formal GLP toxicology study will need to be performed for a complete safety assessment before Ab4B19 proceeds for clinical studies.

An important question that remains to be answered is whether the preclinical discoveries on Ab4B19 are translatable in human POF. Genetic association studies have identified a link between *Bdnf* polymorphism and POF. A reduced level of BDNF protein has been reported in the plasma of menopausal women and POF patients[34,61]. However, previous human studies were fragmented and circumstantial, and there was no evidence that the phenotypes of TrkB knockout mice are translatable. Our single-cell RNA-seq showed relatively high levels of BDNF expression in follicles of all stages of the human ovary, indicating its role throughout follicle development. However, BDNF expression is downregulated in the ovaries of aged women (Fig. 7f, g and Supplementary Fig. 6a, b). Further, the expression pattern for TrkB-FL mRNA, which is similar to that in mice, suggests that BDNF-TrkB may function transiently in primary/secondary follicle stages for GCs but continuously from the secondary follicle stage on for oocytes in humans. Finally, Ab4B19 elicited better effects than BDNF on CREB phosphorylation and cell survival in human-derived KGN cells (Fig. 7a, b). More importantly, Ab4B19 could activate TrkB downstream signaling in human ovary tissue ex vivo (Supplementary Fig. 6c, d). These human data reveal the translational potential of Ab4B19 for POF.

In summary, Ab4B19 not only extends and enhances the reproductive capability in the natural aging mice but also protects the fertility and repairs the developing follicles' damage in the Cy-treated mice. These results identified a potential first-in-class drug for clinical therapy of POF, with differentiated target/mechanism and novel chemical entity.

## Methods

**Animal model.** Female (15 days, 6–8 months, and 6–8 weeks old) and male (6–8 weeks old) C57BL/6 mice were purchased from Beijing Vital River Laboratory Animal Technology Co., Ltd. and housed in the specific pathogen-free (SPF) condition with temperature control (22 ± 1 °C) and humidity control (60 ± 10%) on a 12 h light/12 h dark cycle with ad libitum access to water and regular rodent chow. All the animal protocols were approved by Tsinghua University Animal Care and Use Committee (Protocol number: 17-LB17).

For the experiments with 15-day-old mice, they were randomly treated with Ab4B19 and Normal IgG through tail vein injection for 5 days respectively.

For the natural aging POF (NA-POF) model, female C57BL/6 mice of 6–8 months old were purchased. After acclimation until 12 months old, the mice were divided into two groups randomly and treated with Ab4B19 or Normal IgG (1 mg/kg, once every 4 days) through tail vein injection for 16 days.

For cyclophosphamide (Cy)-induced POF (Cy-POF) model, 6–8 weeks old C57BL/6 female mice were weighed and then treated with a single intraperitoneal injection of Cy (75 mg/kg, 200–300 µl), or an equal volume of saline solution as the control. Seven days later, these mice were divided into three groups (n = 12 each) randomly for three different treatments: control (Ctrl), POF + Normal IgG (Vehicle), POF + Ab4B19 (Ab4B19).

**Cell lines culture and proliferation assay.** KGN cells kindly donated by Dr. Yi-Ming Mu (Chinese PLA General Hospital, Beijing, China) were cultured as previously described, with Dulbecco's Modified Eagle's Medium (DMEM)/F12 medium supplemented with 10% fetal bovine serum (Gibco, USA), 1% penicillin/streptomycin (Gibco, USA). To determine the effects of BDNF and Ab4B19 on KGN cells, Cell Counting Kit-8 (CCK-8) (40203ES76, YEASEN) assay was used to assess cell proliferation. In brief, KGN cells were seeded on 96-well plates. After 24 h incubation in culture medium, BDNF (0.2 nM) or Ab4B19 (0.2, 0.6, or 1.8 nM) was added into wells. After 72 h treatment, 10 µl CCK-8 solution was added into each well and incubated at 37 °C for 2 h. The absorbance at 450 nm was determined using a microplate reader (Biotek, Cytation5). The cell viability was calculated by the optical density (OD) values of treated groups/OD values of the control group × 100%.

**Ovary culture.** Ovaries were dissected from 3-day-old female pups and cultured in 24-well tissue culture plates coated with agarose gel at 37 °C in an atmosphere of a 5% $CO_2$ incubator. Then they were treated with 0.2 nM BDNF or 0.6 nM Ab4B19 for 48 h and harvested for experimental analysis. The culture medium was DMEM/F12 (Gibco, USA) and α-Minimum Essential Medium (MEM, Hyclone, SH30265.01) (1:1) supplemented with 10% fetal bovine serum (Gibco, USA), 1% insulin-transferrin-selenium A mix (Gibco, USA), 1% penicillin/streptomycin (Gibco, USA), 1% sodium pyruvate (Gibco, USA).

**Oocyte in vitro maturation.** Ovaries were isolated from 6–8 weeks old mice and punctured in $M_2$ medium (Sigma, St. Louis, MO, USA). Germinal vesicle intact oocytes were picked by a pipetted tube and cultured in M16 medium [(94.66 mM NaCl, 4.78 mM KCl, 1.71 mM $CaCl_2 \cdot 2H_2O$, 1.19 mM $KH_2PO_4$, 1.19 mM $MgSO_4 \cdot 7H_2O$, 25 mM $NaHCO_3$, 23.28 mM sodium lactate, 0.33 mM sodium pyruvate, 5.56 mM glucose, bovine serum albumin (BSA), 1% penicillin/streptomycin (Gibco, USA)] with 0.2 nM BDNF, or 0.2, 0.6, 1.8 nM Ab4B19 with or without K252a at 37 °C in an atmosphere of 5% $CO_2$ incubator. Liquid paraffin oil was used to cover the medium to prevent evaporation.

**Histological analysis.** Assessment of ovary histology was conducted as previously described[8,15,50]. Briefly, ovaries were collected and fixed in 4% paraformaldehyde (PFA) for at least 12 h. After paraffin-embedding and a serial section for 5 µm thickness, H&E staining and blind follicle counting from every fifth section were performed. The total numbers of various follicles were then multiplied by 5 after counting. Please note that, for the NA-POF model, all sections from an entire ovary were used, and we counted all follicles of different stages in an ovary. Primordial follicles per ovary was counted only when the nucleus was clearly identified. Pre-antral and antral follicles were counted when the nucleus of the oocyte was clearly identified. Different follicle stages were classified as the following. Primordial follicles: an oocyte was surrounded by a single layer of flattened squamous follicular cells. Pre-antral follicles: oocytes with two or more layers of cuboidal granulosa cells, but no antrum. Antral follicles: oocytes and antrum within multiple layers of granulosa cells and a theca layer. Atretic follicles were identified by vacuous follicles with degenerating oocytes and pycnotic granulosa cells. Corpora lutea counting was based on their morphology and diameter. Similar methods were used to count various follicles in the Cy-POF model.

**Immunofluorescence staining.** The isolated ovaries were fixed and permeabilized with 4% PFA and then prepared for paraffin section. After dewaxing and rehydration of paraffin sections, membrane permeation was performed after washing. Then the sections were blocked with blocking buffer (5% goat serum, 0.3% Triton™ X-100 in 0.1 M PBS) for 1 h at room temperature, followed by the incubation with primary antibody overnight at 4 °C. After washing with TBST, sections were incubated with the secondary antibody at 37 °C for 1 h. After washing again, DAPI was added to probe the nuclei and mounted with a mounting medium (H-1400, Vector Laboratories, Inc). Axio Scan.Z1 (Zeiss) and Zeiss LSM780 were used for widefield and confocal imaging, respectively. Images were captured and analyzed using software Zeiss Zen (v2.1) and Image J (v1.52i). The primary antibodies used for immunostaining included: anti-DDX4 (1:200, ab13840, Abcam), anti-AMH (1:400, sc-6886, Santa Cruz Biotechnology), anti-cleaved CASP3 (9664, CST), anti-Foxo3 (1:250, Santa Cruz Biotechnology), and anti-BDNF (1:100, Abcam, ab108319). The secondary antibodies included: Alexa Fluor 546 conjugated goat anti-mouse (Thermo Fisher Scientific, A-11030, 2 µg/ml) and Alexa Fluor 488 conjugated goat anti-rabbit (Thermo Fisher Scientific, A-11008, 2 µg/ml).

**ELISA.** ELISA was conducted as the following: Briefly, TrkB-ECD, the coating protein, was dissolved in coating buffer (0.1 M Carbonate Buffer, pH 9.6, $NaHCO_3$ 8.4 g/L) and used to coat the 96-well ELISA plate (Corning, cat.9018) overnight at 4 °C. After washing and blocking, standard antibody proteins or samples (serum or protein lysates) in suitable dilutions were added into wells and incubated for 2 h. Then the plate was washed again and incubated with HRP-labeled secondary antibody for 30 min. Finally, the absorbance at 450 nm was read by a microplate reader (Biotek, Cytation5). Data were analyzed using the software GEN5 CHS (Biotek, v2.05).

**Serum analysis**. After Ab4B19 administration, the blood samples were collected from the mice orbital sinus under anesthesia with avertin. After clotting at room temperature for 90 min, serum was obtained via centrifuging at $1800 \times g$ and stored at −80 °C for further analysis. The serum levels of E2 and FSH were analyzed using the standard protocols of the ELISA kits (Cloud-clone, Wu Han). The levels of testosterone, high-density lipoprotein (HDL), and low-density lipoprotein (LDL) were determined with [125]I-labeled RIA kits (Beijing North Institute of Biological Technology). BDNF levels in serum collected from mice of different ages were measured using the standard protocol of the ELISA kit (Genstar, C643-01).

**Vaginal smears and estrous cycle determination**. After the second treatment with Ab4B19, estrous cyclicity was monitored by vaginal smears taken daily at the same time each day over at least three consecutive estrous cycles (12 days). In brief, saline solution was expelled into the vagina using the pipette and collected in a 1.5 ml tube, followed by transferring onto a glass slide for microscopic analysis. The estrous cycles were determined by the vaginal cytology of dominant cell type: proestrus stage: round, nucleated epithelial cells; estrus stage: cornified squamous epithelial cells; metestrus stage: epithelial cells and leukocytes; and diestrus stage: nucleated epithelial cells and a predominance of leukocytes.

**Western blotting**. KGN cells, mouse, or human ovarian tissues were lysed with the lysis buffer (20 mM HEPES,150 mM sodium chloride, 1.0% CA-630, 0.1% SDS, 2 mM Ethylenediaminetetraacetic acid, 1% deoxycholic acid, protease, and phosphatase inhibitor mixture (Roche Diagnostics)). Protein concentrations were measured by BCA kit (Thermo Scientific). Thereafter, proteins were denatured at 98 °C for 12 min and resolved by SDS-PAGE (10% or 15% SDS-PAGE gel). After transferring proteins onto PVDF membranes (BIO-RAD) followed by incubation with blocking buffer for 1 h at room temperature, the membranes were incubated with the primary antibodies in blocking solutions at 4 °C overnight before detection with HRP-conjugated secondary antibodies (Cell Signaling Technology, #7074, 1:4000). Blot signals were visualized using SuperSignal™ West Pico Chemiluminescent Substrate (Thermo Scientific) by Tanon 5200 (software: Tanon MP, v1.02) and finally displayed on the gel imager for grayscale analysis using Tanon Gis (v4.2). The following primary antibodies were used: anti-TrkB (Rabbit, CST,1:1000), anti-pTrkB (Rabbit, CST, 1:1000), anti-Akt (Rabbit, Easybio, 1:1000), anti-pAkt (Rabbit, CST, 1:2000), anti-CREB (Rabbit, CST, 1:1000), anti-pCREB (Rabbit, CST, 1:1000), anti-ERK (Rabbit, CST, 1:1000), anti-pERK (Rabbit, CST, 1:2000), anti-DDX4 (ab13840, Abcam, 1:1000), anti-pFOXO3a (9464, CST, 1:1000) anti-BAX (ab32503, Abcam, 1:2000), anti-Bcl-2 (sc-7382, Santa Cruz, 1:800), anti-cleaved CASP3 (9664, CST, 1:1000), anti-AMH (Rabbit, Abcam, 1:1000) anti-FSHR (Rabbit, Bioworld, 1:1000), anti-GAPDH (Mouse, Easybio, 1:3000), anti-β-actin (AA128, Beyotime, 1:1000).

**Superovulation and oocyte morphological analysis**. For the fertility assessment, oocyte superovulation is carried out for all experimental groups, which were treated by an injection (i.p.) of pregnant mare serum gonadotrophin (PMSG, 5UI), followed by the administration (i.p.) of hCG (5UI) 48 h later. Oocyte-cumulus complexes were collected from the ampulla 13–14 h after hCG administration. Oocytes were counted after enzymatic dissociation from the surrounding cumulus with a solution of 10 mg/ml hyaluronidase (Millipore-Sigma). The evaluation of oocyte quality was performed according to the morphological criteria from a previous review. Mouse oocytes are classified into: (1) normal oocytes, (2) abnormal oocytes with extracytoplasmic abnormalities (thick or distorted zona and large perivitelline space), or intracytoplasmic abnormalities (dark or granular cytoplasm and cytoplasmic fragments), or shape abnormalities.

**Fertility assessment**. For the fertility assessment, breeding examination and detection of embryos' implantation sites on pregnant mouse uterus were carried out. Female mice were housed with identifiably fertile C57BL/6 males at a 1:1 ratio. Successful mating was judged by observation of a vaginal plug. For NA-POF mice, quantitative analyses for embryo implantation sites at around pregnancy day 7.5 were used to verify the fertility, and other fertility parameters such as fertility rate and the number of pups were measured 2 months after the therapeutic administration of Ab4B19. The healthy status of offspring mice delivered from the Ab4B19 treatment group was examined and compared with the control mice of the same age. For the Cy-induced POF model, treated females were crossed with fertile males at a 1:1 ratio and examined for the offspring delivery lasting for at least 3.5 months. The number of pups per litter and pup weight at postnatal day 2 were recorded and statistically quantified.

**Transcriptional analysis of the Bdnf and TrkB in human GCs and oocytes**. The raw data of single-cell RNA-seq were derived from the previously established database and similar analysis processing was used[62]. Briefly, the number of genes detected were counted in each cell. Cells with fewer than 2400 genes or 500,000 mapped reads were filtered out. In total, 80 oocytes and 71 GCs at five developmental stages passed the filter standards. To ensure the accuracy of estimated gene expression levels, only genes with fragments per kilobase per million (FPKM) >1 in at least one cell were analyzed. Expression levels of the *Bdnf* or *TrkB* gene were represented by FPKM plus one then log2 transformed in the

following analysis. The data of BDNF, TrkB-FL (full-length TrkB), and TrkB-T1 (Truncated TrkB-1) in GCs and oocytes from five different developmental stages of ovarian follicles were analyzed and performed with the software GraphPad Prism (GraphPad, v7.0).

**Human ovarian sample preparation**. The experiments using human ovarian tissues in this study conformed to the Declaration of Helsinki and the protocol was approved by the Ethics Committee of Beijing Obstetrics and Gynecology Hospital. Participants were compensated for their participation. With written informed consent, fresh ovarian tissues were obtained from seven female donors who underwent ovariectomy or hysterectomy. All the donors ranged in age from 20 to 64 years, with no history of any autoimmune or genetic disease. These donors did not undergo hormonal therapy at least 6 months before surgery and were not exposed to any cytotoxic agents or radiotherapy. No ovary samples exhibited any histopathological abnormality, as was confirmed by gynecological pathologists.

To examine downstream signal activation by TrkB agonistic antibody, human ovarian tissues were dissected from female donors who underwent surgery for endometrial stromal sarcoma, removed of fibrous tissues of the tunica albuginea and some fatty tissues, fragmented into small cubes, and incubated in 48-well tissue culture plates with culture medium at 37 °C in a 5% $CO_2$ incubator. Then they were treated with 1 nM BDNF, 3 nM normal IgG, or 3 nM Ab4B19 for 35 min and harvested for Western blotting analysis. Since the ovarian pieces were mixed and divided into three groups, a similar number of follicles were distributed in each group. The culture medium was DMEM/F12 (Gibco, USA) and α-Minimum Essential Medium (MEM, Hyclone, SH30265.01) (1:1) supplemented with 10% fetal bovine serum (Gibco, USA), 1% penicillin/streptomycin (Gibco, USA).

**Novel object recognition**. The novel object recognition test (NORT) was conducted under dim light (~5 lux) in a plastic open-field box (40 cm length × 40 cm width × 30 cm height). A digital camera was installed above the box to record the behavior of the animals. Mice were habituated to the testing room and the behavioral box at 10 a.m. for 3 consecutive days before testing (30 min for the testing room and 5 min for free exploration of the empty box). During the acquisition phase, a mouse was allowed to freely explore the two identical objects (bottles) for 5 min. During the test phase, one of the two objects was replaced by a novel object (woody brick) and the mouse was allowed to freely explore for 10 min. The amount of time that the mouse spent exploring the novel and familiar objects was counted, respectively. Exploration time and the discrimination ratio of the acquisition phase and test phase were presented. The discrimination ratio was calculated as score = (time spent with novel object − time spent with identical object)/(total time spent with both objects).

**Statistics**. Statistical analyses were performed with the software GraphPad Prism (GraphPad, v7.0). Statistical tests and *P* values are reported in the text and figure legends. Data were presented as mean ± SEM. Significance was defined as *$P < 0.05$, **$P < 0.01$ and ***$P < 0.001$.

**Reporting summary**. Further information on research design is available in the Nature Research Reporting Summary linked to this article.

## Data availability

All relevant data and code are available within the article and its supplementary information/Source data or freely available from the corresponding authors upon reasonable request. RNA-Seq data is from the NCBI Gene Expression Omnibus (GEO) under the accession number: GSE107746. RNA-Seq data for BDNF and TrkB expression (Fig. 7c–e) are also available in Source data. Source data are provided with this paper.

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

## Acknowledgements
We would like to thank Dr. Yi-Ming Mu for the kind gift of the KGN cell lines. We are grateful to Dr. Fu-Dong Shi for advice on the manuscript; and Yihua Xu, Shudan Wang, Fang Han, Yang Dou, Longping Liu, and Jizhou Li for advice and technical support. This work was supported by the National Key Research and Development Program of China (2017YFE0126500 to B.L.), the National Natural Science Foundation of China (81861138013 to B.L., 81501105 to W.G., and 31730034 to B.L.), Beijing Advanced Innovation Center for Human Brain Protection, and Beijing Municipal Science & Technology Commission (Z151100003915118 to B.L.).

## Author contributions
X.Q., W.G., and B.L. initiated the project and designed the study. X.Q. conducted the experiments and analyzed the data. T.Z. contributed to the behavioral tests. Y.Z. and J.Q. provided the single-cell RNA-seq data of human ovarian follicles as well as experimental guidance on study design and safety evaluation. C.Y. provided the human ovarian samples. X.Q., B.L., and W.G. wrote, Y.Z, and J.Q edited the manuscript.

## Competing interests
B.L. and W.G. are co-inventors of the filed patents related to the TrkB agonistic antibodies. B.L. is also a co-founder and Scientific Advisor of 4B Technologies, Limited, a biotech company that develops medicines for neurodegenerative diseases. The remaining authors declare no competing interests.
