## [Peer Review File · Nature Communications]

TrkB agonist antibody ameliorates fertility deficits in aged and cyclophosphamide-induced premature ovarian failure model miceREVIEWER COMMENTS

Reviewer #1 (Remarks to the Author):

The study by Qin et al. addresses an important issue, namely a major cause of female infertility. It delineates a new possible therapeutic approach, based in large part on knowledge accumulated over the last decade as well as on findings reported in the submission.

Briefly, using a mouse model, the study reports on the use of a previously characterised antibody activating the BDNF receptor TrkB to promote oocyte maturation. It concludes that the results “pave a new way for a therapeutic utility” of the antibody in question in the context of premature ovarian failure.

In general, the results support this conclusion, in particular in view of the demonstration that 12-month old mice generate more oocytes and fewer abnormal oocytes when treated with the antibody every 4 days for a period of two weeks. The conclusion is also supported by observations made with mice treated with cyclophosphamide in an attempt to mimic a cause of infertility in humans. Last the potential relevance of the findings to humans is investigated using the human cell line KGN.

Whilst these pre-clinical data are interesting and convincing the overall impression is that the study is more about connecting existing dots than truly novel findings. In particular, the superiority of using TrkB activating antibodies over BDNF as therapeutic agents has already been widely and repeatedly publicised and the usefulness of the antibody used here reported in models of motoneuron disease and ischemic brain injury. In addition, as acknowledged by the authors, a number of previous studies in the mouse as well as in some other species have already detailed the characteristics and the biological significance of the BDNF/TrkB signalling system in the ovaries as well as the beneficial effects of TrkB activation in the context of oocyte maturation.

Additional comments:

1. Some of the Western blot results are unconvincing and the illustration limited to narrow windows (see in particular the Western with Ab4B19, Fig. 1b). As TrkB levels are presumably used to normalise the degree of TrkB activation, itself rather underwhelming, the documentation of these results must be reconsidered.

2. The origin of the RNAseq data is not entirely clear as they seem to derive from another study (Ref. 58). Somewhat surprisingly, reference to these results is made in the Abstract, whilst neither their acquisition nor the full set of data are presented in this submission.

Reviewer #2 (Remarks to the Author):

In this preclinical study, the authors investigated the utility of a specific antibody (Ab4B19) as a potential therapy against premature ovarian failure (POF). Ab4B19 reached ovarian follicles after intravenous injection and activated TrkB signaling and promoted folliculogenesis. Importantly, in chemotherapy induced ovarian failure mouse model, treatment with Ab4B19 reversed most of POF-related phenotype features. The manuscript is focused and well written. The following specific comments are provided.

Specific Comments

(1) Originality: The work is original and important. It is highly translational and has the potential to inform clinical practice.

(2) While the safety data of AB4b19 intravenous delivery seems favorable and reassuring in first and second generation pups. There is data provided on the safety profile in the mother herself and if such pleiotropic protein might have some off target effects.

(3) Figure 4C suggest that hysterotomy was done, was there a difference between implantation site versus number of delivered pups suggesting intrauterine resorption of fetuses?

(4) The discussion did not thoroughly address the challenges of translating this to women with POI which a heterogeneous group, mostly (70%) idiopathic, another 10-15% likely has chemotherapy for several years and now has POI, that could be conceptionally different from the fresh recent onset POI treated in this mice after few days of acute onset disease.

Minor Comments

- (1) The field is moving away from the premature ovarian failure terminology and towards premature ovarian insufficiency.
- (2) Line 43-45: This statement is confusing. POF patients taking contraceptives (? hormonal) should not need additional hormone replacement therapy...Pls rephrase.
- (3) Line 74: remove "in"
- (4) Line 115: replace "granule" with "granulosa"

Reviewer #3 (Remarks to the Author):

POF is common and currently there are no effective treatments. In this study, an agonistic antibody (Ab4B19) for the BDNF receptor, TrkB, was used to treat two mouse models of POF. The idea is that, this antibody stimulates TrkB signalling, which promotes follicle development and growth, or prevents follicle atresia. Confusingly, a case seems to be made for both scenarios, but the mechanisms by which this might occur are not identified. This strategy for fertility restoration/preservation has been investigated for other targets, and thus the concept in itself, is not novel.

Aging mice, or mice treated with cyclophosphamide, were injected intravenously with the antibody. This appeared to stimulate the development of follicles to the antral follicle stage, and improved fertility ageing mice, and to prevent follicle loss/stimulate development in Cy-treated mice BDNF and TrkB expression was detected in human follicles. Human ovarian cancer granulosa cell tumor line KGN, was also studied. The conclusion was that Ab4B19 might be candidate drug for POF therapy.

Overall, this study builds on a significant body of existing data showing that BDNF and TrkB have important roles in follicle development. This study utilizes an antagonistic antibody in place of genetic models or BDNF supplementation, is a small step forward in terms of knowledge gain. From a potential therapeutic point of view, it is not clear from the data presented in here, if such a strategy could/would work in humans. The translational data presented are very minimal. I have a number of serious concerns about the quality and interpretation of the data.

1. Overall, the writing of the introduction should be improved to more clearly and accurately describe the context of the study. Grammar requires considerable attention.
2. This statement in the introduction does not make sense "Clinical care for POF patients who are at menopause age". If women are at the normal age of menopause, then by definition, they are not experiencing POF.
3. "While there is no well-established genetic model for POF". This is a confusing statement. There is unlikely to one single cause of POF, and consistent with this concept there are many genetic mouse models in which POF is a feature.
4. "Unfortunately, targets and mechanisms for these treatments are largely unknown, and none of them could promote folliculogenesis of residual follicles in patients. Thus, these efforts are not very effective." This statement is a misleading and does not appropriately recognise the significant amount of knowledge already available, indeed some important contributions have been ignored. A great deal is now known about the cell death mechanisms responsible for follicle depletion and targets/pathways have been characterised by a number of different groups. Additionally, the PI3K/mTOR pathways has been studied in some detail as a method for controlling follicle activation and promoting follicle development to protect/restore fertility in the context of natural aging or premature follicle depletion.

For example, treatment of ovarian fragments from POI patients with PTEN-inhibiting and AKT-activating drugs promotes follicle growth (Kawamura K, Cheng Y, Suzuki N, et al. Hippo signaling disruption and Akt stimulation of ovarian follicles for infertility treatment. Proc Natl Acad Sci U S A. 2013;110(43):17474-17479.).

5. Supplemental figure 5 is inaccurate. It shows Cy only depleting growing follicles, when it is well established that Cy causes DNA damage in the oocytes of primordial follicles, leading to oocyte apoptosis and depletion of the pool.

6. The data supporting penetration of Ab4B19 into follicles are not very convincing. Line 110 "Immunostaining was performed using a FITC-tagged anti-IgG secondary antibody to detect Ab4B19 in follicles at different time points after its tail-vein injection (1 mg/kg; iv) into adult mice." An anti IgG antibody will detect all IgGs, not just Ab4B19, so this does not confirm that Ab4B19 penetrated follicles, unless the secondary antibody is species specific. Can the authors please clarify the details of the anti-IgG, being sure to note whether it was cross adsorbed against mouse IgG. It is also stated that Supplemental Figure 1A demonstrates penetration of the antibody into the follicular antrum. There are no follicular antrums shown in this figure. Additionally, in Supplemental Figure 1B it looks like background antibody sticking to the zona pellucida. It seems unlikely that antibodies could penetrate the zona pellucida and contact the oocyte, which I believe is where TrkB is expressed. Can the authors please confirm which mouse ovarian cell types TrkB is expressed by and at which stages of development. For Figure 1 A, it would be more convincing to localise Ab4B19 with TrkB.

7. It seems like the in vitro maturation experiments described in Figure 2 using denuded oocytes. If so, this implies that Ab4B19 is having a direct effect on oocytes. But how does Ab4B19 penetrate the zona pellucida to activate the receptor on the oocytes?

8. The follicle counting data in Figure 2F are a little strange. The number of primordial and preantral follicles are similar, but it is well established that primordial follicles are more abundant. I suspect a major methodological error. Follicle counting methodology should be described in more detail- especially for the antral follicles. I am also unsure why pre-pubertal animals were used for this experiment. It would be more relevant to use sexually mature animals as follicles may respond differently.

9. Figure 3D. 8 days is far too short to measure estrus cyclicity accurately or meaningfully in mice, especially with such low numbers of mice

10. Figure 3i. Measurement of E2 using ELISA in mice is widely considered to be very inaccurate and lacking in sensitivity required, especially in reproductively aged mice.

11. Figure 4, the total number of oocyte ovulated should also be shown, not just the % abnormal. How many pups were analysed in part F? I am not convinced fertility studies on 8 mice per group is sufficient as fertility can vary significantly from one animal to the next in aged mice. Data from individual mice should be shown as dots on each of the bar graphs (this comment applies to all data presented in the manuscript).

12. Mice are treated with 75mg/kg cyclophosphamide to generate "Cy-POF" mice and follicles were counted 13 days after this. I am not convinced this is a good model for the purposes of this study. The primordial follicles were only partially depleted by 75mg/kg Cy and this is unlikely to induce POF. Studies have shown that even after 150mg/kg cyclophosphamide, mice are fertile, and can produce quite a few litters of normal size. Partial depletion of primordial follicles is not generally associated with depletion of growing follicles, as there is sufficient primordial follicles to restore the growing follicle pool. So, it is unclear why there is a reduction in growing follicles- unless the authors are claiming that Ab4B19 prevents atresia of primordial AND growing follicles. What are the mechanisms by which Ab4B19 would attenuate follicle loss at all stages of development? Cy -induced follicle loss is known to be caused by DNA damage to oocytes and granulosa cells and the induction of the intrinsic apoptosis pathway- how Ab4B19 could prevent this is not clear. Changes in the expression of

Bcl2 and Bax, or caspase 3, were measured too long after Cy treatment to be meaningful. Most effects of Cy occur with the first 24-48 hours of treatment. How could Ab4B19 given a number of days after this Cy induced follicle depletion then restore primordial follicle numbers- this is impossible unless the authors are claiming stimulation of an oogonial stem cell- extremely controversial. The arguments made in the discussion regarding this are not convincing. Others have shown that even with higher doses of Cy, litter size is normal, but the authors show a reduction here. How do this authors explain this discrepancy?

Reviewer #1 (Remarks to the Author):

The study by Qin et al. addresses an important issue, namely a major cause of female infertility. It delineates a new possible therapeutic approach, based in large part on knowledge accumulated over the last decade as well as on findings reported in the submission. Briefly, using a mouse model, the study reports on the use of a previously characterised antibody activating the BDNF receptor TrkB to promote oocyte maturation. It concludes that the results “pave a new way for a therapeutic utility” of the antibody in question in the context of premature ovarian failure. In general, the results support this conclusion, in particular in view of the demonstration that 12-month old mice generate more oocytes and fewer abnormal oocytes when treated with the antibody every 4 days for a period of two weeks. The conclusion is also supported by observations made with mice treated with cyclophosphamide in an attempt to mimic a cause of infertility in humans. Last the potential relevance of the findings to humans is investigated using the human cell line KGN.

Whilst these pre-clinical data are interesting and convincing the overall impression is that the study is more about connecting existing dots than truly novel findings. In particular, the superiority of using TrkB activating antibodies over BDNF as therapeutic agents has already been widely and repeatedly publicised and the usefulness of the antibody used here reported in models of motoneuron disease and ischemic brain injury. In addition, as acknowledged by the authors, a number of previous studies in the mouse as well as in some other species have already detailed the characteristics and the biological significance of the BDNF/TrkB signalling system in the ovaries as well as the beneficial effects of TrkB activation in the context of oocyte maturation.

Response: We appreciate the reviewer’s overall positive comments, but after seeing our revised manuscript, we hope that the reviewer could be persuaded that our work is not just “connecting existing dots”, but a significant step forward in translating decades of neurotrophin research into something of real clinical values. For the first time, we have identified a potential first-in-class drug for clinical therapy of POF, with novel target/mechanism and novel chemical entity.

First, although preclinical (BDNF regulation of folliculogenesis) and clinical (decrease in plasma BDNF level in POF patients, and GWAS identification of *Bdnf* as a POF-risk gene) evidence exists for a role of BDNF in POF, no disease-modifying drug has ever been developed for POF. Our work has laid a foundation for a truly innovative therapy for POF. All previous clinical trial using BDNF as a therapeutic agent have failed. It has now been appreciated that BDNF itself cannot be a drug, due to its three major limitations (poor PK, poor diffusibility, activation of p75^{NTR}). Indeed, we have now shown that in the Cy-POF model, treatment with Ab4B19, but NOT BDNF, markedly attenuated the Cy-induced gonadotoxicity and therefore rescued the number and quality of ovulated oocytes in Cy-POF ovaries (Supplementary Fig. 5). Our study provides compelling evidence that TrkB could serve as a novel target for POF with differentiated mechanisms (not activating p75^{NTR}), and confirms superiority of our

TrkB activating antibody over BDNF as therapeutic agents for the treatment of POF, a truly unmet medical need.

Second, many TrkB agonists have been developed, with the aim to mimic the biological functions of BDNF and overcome its limitations, in the treatment of nervous system diseases. However, in all the patents published so far for TrkB agonists or antibodies, POF has never been listed as a disease indication. Therefore, the present study describes the first chemical entity for the treatment of POF using TrkB as a drug target. Our work is also the first interdisciplinary translational study that combines insights from the neurotrophin field and reproductive biology.

Finally, previous human studies were fragmented and circumstantial, and there was no evidence that the phenotypes of TrkB knockout mice are translatable. Our team carried out analysis of BDNF and TrkB expression in human ovary at single-cell level, revealing a time window in which TrkB drug treatment might be effective. Moreover, our new data suggest that BDNF expression is down-regulated in the ovaries of older women (Fig. 7f, g and Supplementary Fig. 6a) and Ab4B19 could activate TrkB downstream signaling in human ovary tissue *ex vivo* (supplementary Fig. 6b, c). These results reveal the translational potential of the present study: treatment of POF patients with a first-in-class drug - TrkB antibody.

Additional comments:

1. Some of the Western blot results are unconvincing and the illustration limited to narrow windows (see in particular the Western with Ab4B19, Fig. 1b). As TrkB levels are presumably used to normalise the degree of TrkB activation, itself rather underwhelming, the documentation of these results must be reconsidered.

Answer: We have now repeated the experiment of Fig. 1b. The new immunoblots are shown.

2. The origin of the RNAseq data is not entirely clear as they seem to derive from another study (Ref. 58). Somewhat surprisingly, reference to these results is made in the Abstract, whilst neither their acquisition nor the full set of data are presented in this submission.

Answer: The original experiment of single-cell RNAseq was done by members of our team and published before (original Ref. 58, now Ref. 54). It contains a large collection of original, unprocessed data. In this study, we analyzed the relevant raw data to assess the expression levels of BDNF and TrkB in oocytes and granulosa cells. In the revised manuscript, we have described the detailed methods for data acquisition and analysis (in Methods), provided the original data relevant to BDNF and TrkB (in Supplementary Table 7), in addition to the results and conclusion derived from these original data (in Results).

Reviewer #2 (Remarks to the Author):

In this preclinical study, the authors investigated the utility of a specific antibody

(Ab4B19) as a potential therapy against premature ovarian failure (POF). Ab4B19 reached ovarian follicles after intravenous injection and activated TrkB signaling and promoted folliculogenesis. Importantly, in chemotherapy induced ovarian failure mouse model, treatment with Ab4B19 reversed most of POF-related phenotype features. The manuscript is focused and well written. The following specific comments are provided.

Response: We thank the reviewer for his/her positive comments on our manuscript. A point-to-point response to the reviewer's comments is as following.

Specific Comments

(1) Originality: The work is original and important. It is highly translational and has the potential to inform clinical practice.

Answer: We greatly appreciate the reviewer's laudatory remarks.

(2) While the safety data of AB4b19 intravenous delivery seems favorable and reassuring in first and second generation pups. There is (no?) data provided on the safety profile in the mother herself and if such pleiotropic protein might have some off target effects.

Answer: We have now included the blood test results for the mother mice (supplementary Table 5). There are altogether 6 items and none of them were changed in female mice treated with Ab4B19. Moreover, our previous work also demonstrated that repeated dosing of a TrkB agonistic antibody for 9 months had no obvious off-target effect. There was a small reduction of body weight, as well as an increase in lifespan, which are the expected benefits of activation of BDNF-TrkB pathway (Wang et al., 2020).

(3) Figure 4C suggest that hysterotomy was done, was there a difference between implantation site versus number of delivered pups suggesting intrauterine resorption of fetuses?

Answer: Yes, we performed hysterotomy and examined embryo implantation. Our analyses of embryo implantation and litter size indicate that the number of actual birth was generally lower than the number of embryos implanted in the uterus, due most likely to intrauterine abortion or absorption of embryos. In addition, old female mice often display infanticide behavior or consuming the bodies of their own offspring because the pups are often too weak or sometimes die soon after birth. To minimize these problems, we also paid particular attention to the old pregnant mice around the time they gave birth. Thus, we performed statistical analysis of both litter size and embryo implantation number for the NA-POF model, whereas for the Cy-POF model, only litter size counting was sufficient.

(4) The discussion did not thoroughly address the challenges of translating this to women with POI which a heterogeneous group, mostly (70%) idiopathic, another 10-15% likely has chemotherapy for several years and now has POI, that could be conceptionally different from the fresh recent onset POI treated in this mice after few

days of acute onset disease.

Answer: We thank the reviewer for this important comment. The animal models in our study could not represent all types of human POF/POI, which is a heterogeneous disorder with 70% idiopathic. Moreover, POF/POI patients who had been through chemotherapy (10-15%) for several years may also be different from Cy-POF/POI mice with acute onset. It would be interesting to determine the effect of Ab4B19 on mice that had been treated with Cy for different lengths of time. The main goal of the present study was to demonstrate the feasibility of our therapeutic strategy, which is to promote the residual follicular development and maturation with the TrkB agonistic antibody, and protect damaged follicles (such as Cy induced POF/POI). Our results imply that TrkB antibody can be used to treat different types of POF/POI patients regardless of their etiology or onset time, as long as they have residual follicles. Whether this strategy is truly effective in certain subtypes of POF/POI patients will need to be tested through clinical trials.

We have now added these statements in the “Discussion”.

Minor Comments

(1) The field is moving away from the premature ovarian failure terminology and towards premature ovarian insufficiency.

Answer: We appreciate the suggestions. Given that many in the field still used the term POF, we have taken a compromised approach, and explained and changed the term accordingly.

(2) Line 43-45: This statement is confusing. POF patients taking contraceptives (? hormonal) should not need additional hormone replacement therapy...Pls rephrase.

Answer: We thank the reviewer for pointing out this inaccurate expression, and have now rephrased it to “Clinical care for women with POF involves hormone replacement therapy (HRT) plus psychological support or taking contraceptive drugs for those who are not desiring pregnancy”.

(3) Line 74: remove “in”

Answer: We thank the reviewer for pointing out this error, and have now corrected it.

(4) Line 115: replace “granule” with “granulosa

Answer: We thank the reviewer for pointing out this error, and have now corrected it.

Reviewer #3 (Remarks to the Author):

POF is common and currently there are no effective treatments. In this study, an agonistic antibody (Ab4B19) for the BDNF receptor, TrkB, was used to treat two mouse models of POF. The idea is that, this antibody stimulates TrkB signalling, which promotes follicle development and growth, or prevents follicle atresia. Confusingly, a case seems to be made for both scenarios, but the mechanisms by which this might

occur are not identified. This strategy for fertility restoration/preservation has been investigated for other targets, and thus the concept in itself, is not novel.

Aging mice, or mice treated with cyclophosphamide, were injected intravenously with the antibody. This appeared to stimulate the development of follicles to the antral follicle stage, and improved fertility ageing mice, and to prevent follicle loss/stimulate development in Cy-treated mice. BDNF and TrkB expression was detected in human follicles. Human ovarian cancer granulosa cell tumor line KGN, was also studied. The conclusion was that Ab4B19 might be candidate drug for POF therapy.

Overall, this study builds on a significant body of existing data showing that BDNF and TrkB have important roles in follicle development. This study utilizes an antagonistic antibody in place of genetic models or BDNF supplementation, is a small step forward in terms of knowledge gain. From a potential therapeutic point of view, it is not clear from the data presented in here, if such a strategy could/would work in humans. The translational data presented are very minimal. I have a number of serious concerns about the quality and interpretation of the data.

Response: We thank the reviewer's critical review and constructive comments and suggestions, which have helped to improve the manuscript. The reviewer raised a concern on the conceptual novelty of the manuscript. However, we believe that the main thrust of our work is to integrate knowledge and progress in different fields (role of BDNF in ovary, antibody therapy and the biology of follicle development), and to develop a holistic approach to the treatment of a huge unmet medical need – POF. It is a significant advance in translating decades of neurotrophin research into something of real clinical values. In the following, we explain why we believe this might be a potential first-in-class drug for clinical therapy of POF, with novel target/mechanism and novel chemical entity.

First, we believe that the effects on follicle development and the prevention of follicle atresia are two aspects of the same signaling mechanism by BDNF-TrkB used in the treatment of POF. They are not contradictory. TrkB activation can promote follicle development and maturation, and thus reduce the number of atretic follicles. Although the concept of promoting follicle development and growth for the treatment of POF has been mentioned before, TrkB as a drug target has never been realized until now.

Second, while evidence exists on the role of BDNF-TrkB signaling pathway in ovary, especially in follicular development, no pharmaceutical research has been conducted on TrkB as a drug target for POF treatment. In all the patents published so far, POF has never been mentioned as an indication that could be treated by TrkB agonists including TrkB agonist antibody. We have now demonstrated that Ab4B19, an agonistic antibody capable of activating BDNF receptor TrkB, could elicit therapeutic effects that could not be elicited by BDNF itself (Supplementary Fig. 5). Therefore, our work is a critical step forward developing TrkB antibody as a therapeutic agent for POF.

Finally, we have done our best to add more translational data in human. Our new evidence indicated that BDNF expression is down-regulated in the ovaries of older

women (Fig. 7f, g and supplementary Fig. 6a). Further, Ab4B19 could activate TrkB downstream signals in human ovary tissues *ex vivo* (supplementary Fig. 6b, c). Both availability and ethical concerns make it impossible to obtain human oocytes (even from ovarian cancer patients) for functional test. Finally, we are planning to perform similar experiments in large animals such as aged monkeys and pigs, but these will take more resources and much longer time, and therefore are beyond the scope of the present study. We hope the reviewer would agree with us that the full translatability of the TrkB agonistic antibody for POF will need to be tested in human clinical trial, which we are prepared to do.

1. Overall, the writing of the introduction should be improved to more clearly and accurately describe the context of the study. Grammar requires considerable attention.
Answer: Thanks for the suggestions. We have now described the context more accurately, checked grammar by an English writer, and carefully revised “Introduction” accordingly.

2. This statement in the introduction does not make sense “Clinical care for POF patients who are at menopause age”. If women are at the normal age of menopause, then by definition, they are not experiencing POF.

Answer: We have now corrected the misrepresentation accordingly. The sentence has been changed to “Clinical care for women with POF involves hormone replacement therapy (HRT) plus psychological support, or taking contraceptive drugs for those who are not desiring pregnancy.”

3. “While there is no well-established genetic model for POF”. This is a confusing statement. There is unlikely to be one single cause of POF, and consistent with this concept there are many genetic mouse models in which POF is a feature.

Answer: We agree that POF is heterogeneous in nature, and have now corrected the expression accordingly. A good animal model usually requires Face (human disease phenotypes), Construct (disease mechanisms) and Predictive (predict effectiveness of therapy) validities. We mean that there is no good genetic model that could be used for drug efficacy studies (predictive validity). Although several genetic models have been reported to have POF-like phenotypes (including the TrkB knockout mice), they do not involve a general, disease-relevant pathological pathway (such as A β cascade in AD). Therefore, these models may help us to gain insights into the mechanisms underlying human POF, but are not suitable for drug testing.

4. “Unfortunately, targets and mechanisms for these treatments are largely unknown, and none of them could promote folliculogenesis of residual follicles in patients. Thus, these efforts are not very effective.” This statement is misleading and does not appropriately recognise the significant amount of knowledge already available, indeed some important contributions have been ignored. A great deal is now known about the cell death mechanisms responsible for follicle depletion and targets/pathways have been characterised by a number of different groups.

Additionally, the PI3K/mTOR pathways has been studied in some detail as a method for controlling follicle activation and promoting follicle development to protect/restore fertility in the context of natural aging or premature follicle depletion. For example, treatment of ovarian fragments from POI patients with PTEN-inhibiting and AKT-activating drugs promotes follicle growth (Kawamura K, Cheng Y, Suzuki N, et al. Hippo signaling disruption and Akt stimulation of ovarian follicles for infertility treatment. Proc Natl Acad Sci U S A. 2013;110(43):17474-17479.).

Answer: We thank the reviewer for pointing out the inaccurate statement in “Introduction”. What we meant was that targets and mechanisms for “potential protective agents (such as melatonin)”, NOT “cell death mechanisms responsible for follicle depletion”, are largely unknown. We have now revised statement to improve the accuracy of the expression and to recognize the important contributions made by others in revised manuscript.

The study about PI3K/mTOR pathways in controlling follicle activation and promoting follicle development is very important, and the relevant therapeutics is also interesting. However, the treatment process involved surgical removal of a piece of ovary, in vitro culture of ovarian cubes with Akt stimulator to activate the primordial follicles, transplant the activated ovarian cubes back to ovary, and finally retrieval of mature eggs for IVF procedures (see Figure below). This study is very interesting scientifically, but the procedure is a highly complicated and therefore its practical utility may be limited. We have now included this study in “Introduction”.

5. Supplemental figure 5 is inaccurate. It shows Cy only depleting growing follicles, when it is well established that Cy causes DNA damage in the oocytes of primordial follicles, leading to oocyte apoptosis and depletion of the pool.

Answer: The reviewer was right that our description may not be accurate enough. Indeed our data also indicate Cy causes damage and reduction of primordial follicles, which was partially rescued by Ab4B19 treatment. We have now revised the figure and discussed the potential mechanisms in “Discussion”.

6. The data supporting penetration of Ab4B19 into follicles are not very convincing. Line 110 An anti IgG antibody will detect all IgGs, not just Ab4B19, so this does not confirm that Ab4B19 penetrated follicles, unless the secondary antibody is species specific.

Can the authors please clarify the details of the anti-IgG, being sure to note whether it was cross adsorbed against mouse IgG. It is also stated that Supplemental Figure 1A demonstrates penetration of the antibody into the follicular antrum. There are no follicular antrums shown in this figure.

Additionally, in Supplemental Figure 1B it looks like background antibody sticking to the zona pellucida. It seems unlikely that antibodies could penetrate the zona pellucida and contact the oocyte, which I believe is where TrkB is expressed. Can the authors please confirm which mouse ovarian cell types TrkB is expressed by and at which stages of development. For Figure 1 A, it would be more convincing to localise Ab4B19 with TrkB.

Answer: The reviewer was right that “the secondary antibody is species specific”. We apologize for not explaining well. Ab4B19 is a rabbit monoclonal IgG. Therefore, we used anti-rabbit IgG (FITC, absorbed by the IgG from mouse and other species) to detect the localization of Ab4B19.

Also we have changed the statement: “Immunostaining images showing Ab4B19 penetration into follicular antrum 24 hours (24h) after tail-vein injection” to “Immunostaining images showing Ab4B19 penetration into follicles 24 hours (24h) after tail-vein injection” and added an image showing the follicular antrum in Supplemental Figure 1b.

It has been reported that TrkB is expressed in both oocytes and granulosa cells of almost all development stages (Chang et al., 2019; Chow et al., 2020). For granulosa cells, the peak level of relative TrkB expression is in preantral follicles. As for oocytes, the peak level is in antral follicles. And the expression pattern is similar to those of human (Fig. 5d, e). We have tried to use several other TrkB antibodies to co-staining with Ab4B19. Unfortunately, these antibodies seem to compete with Ab4B19, because they are all against to TrkB extracellular domain (no more commercial mouse TrkB antibodies are available for co-staining with Ab4B19). However, in fact, the affinity and specificity of Ab4B19 to TrkB have been well characterized in our previous work (Guo et al., 2019). More importantly, we have now shown more immunostaining images that indicate Ab4B19 not only binds to the surface of oocytes but also endocytoses into cytoplasm (Supplemental Fig. 1b). These results can also be confirmed by the *in vitro* experiments for Question 7 (Supplemental Fig. 1c). More details are discussed below.

7. It seems like the *in vitro* maturation experiments described in Figure 2 using denuded oocytes. If so, this implies that Ab4B19 is having a direct effect on oocytes. But how does Ab4B19 penetrate the zona pellucida to activate the receptor on the oocytes?

Answer: It has been reported that the zona pellucida is permeable to biomacromolecules up to a molecular weight range of 170 KD (IgG is about 150 KD) (Legge, 1995). In addition, we performed a new experiment to determine the penetration of Ab4B19 across the zona pellucida *in vitro* (Supplementary Fig. 1c). We found that Ab4B19 not only penetrates the zona pellucida and binds to the surface of oocytes but also activates TrkB and endocytoses into cytoplasm (TrkB endocytosis depends on its activation).

8. The follicle counting data in Figure 2F are a little strange. The number of primordial and preantral follicles are similar, but it is well established that primordial follicles are more abundant. I suspect a major methodological error. Follicle counting methodology should be described in more detail- especially for the antral follicles.

Answer: We thank the reviewer for pointing out this error. We counted every one out of 5 stained sections, but forgot to time the number by 5. We have now corrected this in “Methods”.

I am also unsure why pre-pubertal animals were used for this experiment. It would be more relevant to use sexually mature animals as follicles may respond differently.

Answer: We have examined the effects of Ab4B19 in both pre-pubertal and sexually mature animals. The main purpose of using pre-pubertal mice was to demonstrate that Ab4B19 is capable of promoting pre-antral follicle growth. The ovaries of pre-pubertal mice contain primarily pre-antral follicles. Thus, the effect of Ab4B19 can be easily determined by counting. We also treated adult mice with Ab4B19 (Fig. 2d, right) to determine whether Ab4B19 can promote the development and maturity of antral follicles

9. Figure 3D. 8 days is far too short to measure estrus cyclicity accurately or meaningfully in mice, especially with such low numbers of mice

Answer: We appreciate the suggestions, and have now re-done the experiments. After 4-day treatment of Ab4B19, estrus cyclicity of the female mice was measured for 12 days, until they were caged with males for mating (Fig. 3a). Male and female were now in the same cages at this time point onward, making it impossible to measure estrus cycle. The number of mice used has been increased to 8. The new data is now presented in new Fig. 3d.

10. Figure 3i. Measurement of E2 using ELISA in mice is widely considered to be very inaccurate and lacking in sensitivity required, especially in reproductively aged mice.

Answer: ELISA has been the method of choice by many to measure E2 (Qi et al., 2019; Tata et al., 2018; Yuan et al., 2016), even though its sensitivity is lower than radioimmunoassay. The level change of E2 in this experiment is highly significant. Thus, we believe the measurement of E2 by ELISA is acceptable. In addition to E2 level, the normalization of hormone is also supported by the upregulated level of AMH and FSHR.

11. Figure 4, the total number of oocyte ovulated should also be shown, not just the % abnormal.

How many pups were analysed in part F? I am not convinced fertility studies on 8 mice per group is sufficient as fertility can vary significantly from one animal to the next in aged mice. Data from individual mice should be shown as dots on each of the bar graphs (this comment applies to all data presented in the manuscript).

Answer: The total number of oocytes ovulated for Fig. 4b has been shown in Fig. 3c (Ctrl and 1 mg/kg Ab4B19 treated groups). This correlation has been mentioned in "Results". For the fertility studies of aged mice, we have now increased the number of mice to 10. In fact, the aged mice we used were 12-14 months old and their fertility was uniformly low. Therefore, the therapeutic effects were highly consistent and significant with 8-10 mice used. Following the instructions on data availability by *Nature Communication*, all data points have been shown for plots.

12. Mice are treated with 75mg/kg cyclophosphamide to generate "Cy-POF" mice and follicles were counted 13 days after this. I am not convinced this is a good model for the purposes of this study. The primordial follicles were only partially depleted by 75mg/kg Cy and this is unlikely to induce POF. Studies have shown that even after

150mg/kg cyclophosphamide, mice are fertile, and can produce quite a few litters of normal size. Partial depletion of primordial follicles is not generally associated with depletion of growing follicles, as there is sufficient primordial follicles to restore the growing follicle pool. So, it is unclear why there is a reduction in growing follicles- unless the authors are claiming that Ab4B19 prevents atresia of primordial AND growing follicles. What are the mechanisms by which Ab4B19 would attenuate follicle loss at all stages of development? Cy -induced follicle loss is known to be caused by DNA damage to oocytes and granulosa cells and the induction of the intrinsic apoptosis pathway- how Ab4B19 could prevent this is not clear. Changes in the expression of Bcl2 and Bax, or caspase 3, were measured too long after Cy treatment to be meaningful. Most effects of Cy occur within the first 24-48 hours of treatment. How could Ab4B19 given a number of days after this Cy induced follicle depletion then restore primordial follicle numbers- this is impossible unless the authors are claiming stimulation of an oogonial stem cell- extremely controversial. The arguments made in the discussion regarding this are not convincing. Others have shown that even with higher doses of Cy, litter size is normal, but the authors show a reduction here. How do these authors explain this discrepancy?

The reviewer's question includes the following major points, which we will answer one by one:

1). Reviewer mentioned that “the primordial follicles were only partially depleted by 75mg/kg Cy and this is unlikely to induce POF”, and suggested that the partial loss of primordial follicles did not affect the number of growth of follicles. Studies have shown that even after 150mg/kg cyclophosphamide, mice are fertile, and can produce quite a few litters of normal size.

Answer: While the results mentioned by the reviewer (which we are not aware of) are possible, it is unclear whether the drug (Cy) administered has indeed reached the target tissues at certain concentrations, and elicited its toxic effects in those experiments. In our hand, 75 mg/kg Cy treatment induced acute (1-2 weeks after treatment) damage (apoptosis, Fig. 5d-f) as well as chronic (more than 3 months) POF-like damage to the ovary (Fig. 6i).

First, the acute injury of follicles caused by Cy was indicated by the reduced number of various follicles, and it affected the reproductive capability. Similar result was presented in the Table I of the paper by Meiorow et al in 1999, showing that the fertility of mice treated by 75 mg/kg Cy decreased significantly after 1 week, began to recover after 2 weeks, and was completely normal after 4 weeks. In the paper by Pascuali et al in 2018, Figure 1A also showed that 75 mg/kg Cy caused significant loss of various follicles after 2 weeks (Meiorow et al., 1999a; Pascuali et al., 2018).

Second, although the number of primordial follicles was decreased, the number of growth follicles remained unchanged during the period of 4-8 weeks after Cy induction. Therefore, the literature showed that the fertility (number of pregnant follicles or litter) of mice did not decrease during this period (Meiorow et al., 1999a; Pascuali et al., 2018). However, our results (round II mating in Fig. 6i) and the literature (Pascuali et al., 2018)

showed that 75 mg/kg Cy treatment induced a significant decrease in fertility after 3.5 months and 12 weeks respectively, indicating that in this moment the primordial follicles were depleted compared with the control. All these results indicated that 75 mg/kg Cy could effectively induce POF.

Table I. Comparison of reproductive performance, as measured by mating rate, number of corpora lutea (CL) and pregnancies, in control and treated animals mated at 4-weekly intervals. Statistical significance as compared with control values is indicated by * ($P = 0.001$). Other data were not significantly different from control values

	Control	Week 1	Week 2	Week 3	Week 4
No. of animals	34	27	29	30	15
No. (%) of animals without CL	12 (35)	10 (37)	9 (31)	12 (40)	5 (33)
Mean no. of CL in mated females	10.4 ± 0.6	8.1 ± 1.0	11.2 ± 0.4	9.6 ± 0.8	10.4 ± 0.7
Mean no. of pregnancy sacs/pregnant animal	8.3 ± 0.8	4.8 ± 1.0*	7.4 ± 0.9	7.0 ± 0.8	7.9 ± 1.2
No. (%) of pregnant animals	21 (61)	12 (44)	18 (62)	17 (56)	9 (60)

Meirow et al., 1999

Table I Cyclophosphamide (Cy) reduces pregnancies and litter size in female mice, which is rescued by Ceramide-1-phosphate (C1P) administration.

	Round 1 (8 weeks after treatment)			Round 2 (12 weeks after treatment)		
	Control	Cy	Cy + C1P 1 mM	Control	Cy	Cy + C1P 1 mM
Mating index	5/5 (100%)	5/5 (100%)	6/6 (100%)	5/5 (100%)	5/5 (100%)	6/6 (100%)
Fertility index (fraction of females that delivered offspring/total) [†]	5/5 (100%)	4/5 (80%)	5/6 (83.3%)	5/5 (100%) ^a	2/5 (40%) ^b	6/6 (100%) ^a
Mean litter size (number of pups) [#]	9.8 ± 0.6	9.6 ± 1.3	8.2 ± 1.1	11.7 ± 1.2 ^a	3.5 ± 1.5 ^b	8.4 ± 1.5 ^{ab}
Mean body weight of pups (g) [#]	1.52 ± 0.09	1.51 ± 0.03	1.64 ± 0.11	1.44 ± 0.07	1.52 ± 0.06	1.62 ± 0.12

Summary of fertility parameters for all the experimental groups (control, Cy and Cy + C1P) and times (8 and 12 weeks post-treatment). Different letters represent statistically significant differences between groups (a vs b: $P < 0.05$), whereas means that share the same letter do not differ significantly. [†]Statistical analysis was performed using Chi-square tests. [#]Statistical analyses were performed by one-way ANOVA followed by Tukey's multiple-comparison tests.

Pascuali et al., 2018

2). Partial depletion of primordial follicles is not generally associated with depletion of growing follicles, as there is sufficient primordial follicles to restore the growing follicle pool. So, it is unclear why there is a reduction in growing follicles- unless the authors are claiming that Ab4B19 prevents atresia of primordial AND growing follicles. **Answer:** Cy treatment causes a decrease in primordial follicles and damages the growing follicles. We believe that the cell damage caused by Cy as a chemotherapeutic agent will last for a period of time in this model. Although the apoptosis induced by Cy through DNA damage generally occurs within 2 days, the damage to the granulosa cells will have a long-term, secondary impact on the growing follicles, leading to their atresia and subsequent cell apoptosis within the follicles. Therefore, the number of growing follicles as well as follicles at other stages would continue to decrease for a long period of time after Cy administration. This is perhaps the reason why apoptotic signals could be detected after about 2 weeks. Similar results were reported by others (Pascuali et al., 2018).

3) What are the mechanisms by which Ab4B19 would attenuate follicle loss at all stages of development? Cy -induced follicle loss is known to be caused by DNA damage to oocytes and granulosa cells and the induction of the intrinsic apoptosis pathway- how Ab4B19 could prevent this is not clear.

Answer: We've already mentioned above that TrkB is expressed in both oocytes and granulosa cells of almost all development stages. Further, the Akt and Erk signaling

pathway downstream of TrkB can promote cell survival and proliferation in a number of system including ovary, which is commonly known (Chang et al., 2019; Chao, 2003) and shown in our manuscript (Fig. 1b, Fig. 7a). The BDNF-TrkB pathway has also been shown to inhibit DNA damage induced apoptosis not only by traditional anti-apoptotic function of Akt signaling but also through synthesis of DNA repair proteins by stimulating CREB (Chao, 2003; Yang et al., 2014), and we have shown that in ovary too (Fig. 5d, Fig. 7b).

4) Changes in the expression of Bcl2 and Bax, or caspase 3, were measured too long after Cy treatment to be meaningful. Most effects of Cy occur with the first 24-48 hours of treatment. How could Ab4B19 given a number of days after this Cy induced follicle depletion then restore primordial follicle numbers- this is impossible unless the authors are claiming stimulation of an oogonial stem cell- extremely controversial.

Answer: In order to demonstrate that Ab4B19 is also a therapeutic drug (rather than preventive/protective), we administered it 7 days after Cy induction. Our data showed that Ab4B19 could inhibit apoptosis more than a week after Cy administration (Fig. 5d) and promote granulosa cell proliferation (Fig. 7a, KGN cells), and therefore protecting follicles at all stages. AMH level was reversed by rescued growing follicles, and thus inhibited the excessive depletion of the primordial follicle pool. In addition, because Ab4B19 was administered 7 days after the onset of Cy-induced injury, we found that the number of follicles was partially rescued, not fully recovered. These have been further elaborated in the "Discussion".

5). Others have shown that even with higher doses of Cy, litter size is normal, but the authors show a reduction here. How do this authors explain this discrepancy?

Answer: This has been discussed in the point 1) above. According to the previous reports, 75 mg/kg Cy induction led to a decrease of fertility in mice due to the decreased growth follicles in the short term, and then the litter size returned to normal due to the continuous supply by the remaining primordial follicles. In a longer term, the fertility of mice would decline again due to a gradual exhaustion of primordial follicles. As shown in the Table I of the paper by Meirou et al. in 1999(Meirou et al., 1999b), the litter size of female mice (5-6 week-old, BALB/c) treated with 75 mg/kg Cy decreased by 42% when the females were mated 1 week after Cy induction, and recovered to 11% and 16% if they were mated 2 and 3 weeks later. The fertility of Cy-treated mice was fully recovered if they were mated 4 weeks after Cy-induction. On the other hand, the litter size of female mice (6-8 week-old C57BL/6) was normal in the 8th week after 75 mg/kg Cy treatment, but decreased by 64% if the females were mated at 12th week, possibly explained by the exhaustion of primordial follicles (see Table I of Pascuali et al. 2018)(Pascuali et al., 2018). Our own results showed that the litter size of female mice (6-8 week-old C57BL/6) decreased by 36% in the first round of mating (13 days after Cy treatment), but further to 25% in the second round (3.5 months after Cy treatment). In general, taken into consideration of the differences in age and stain of the animals used, the changes of litter size we show are highly consistent with those reported in the literatures. We have now explained these in the Results and Discussion.

Reference:

- Chang, H.M., Wu, H.C., Sun, Z.G., Lian, F., and Leung, P.C.K. (2019). Neurotrophins and glial cell line-derived neurotrophic factor in the ovary: physiological and pathophysiological implications. *Hum Reprod Update* 25, 224-242.
- Chao, M.V. (2003). Neurotrophins and their receptors: a convergence point for many signalling pathways. *Nature reviews Neuroscience* 4, 299-309.
- Chow, R., Wessels, J.M., and Foster, W.G. (2020). Brain-derived neurotrophic factor (BDNF) expression and function in the mammalian reproductive Tract. *Hum Reprod Update* 26, 545-564.
- Guo, W., Pang, K., Chen, Y., Wang, S., Li, H., Xu, Y., Han, F., Yao, H., Liu, H., Lopes-Rodrigues, V., *et al.* (2019). TrkB agonistic antibodies superior to BDNF: Utility in treating motoneuron degeneration. *Neurobiology of disease* 132, 104590.
- Legge, M. (1995). Oocyte and zygote zona pellucida permeability to macromolecules. *The Journal of experimental zoology* 271, 145-150.
- Meirow, D., Lewis, H., Nugent, D., and Epstein, M. (1999a). Subclinical depletion of primordial follicular reserve in mice treated with cyclophosphamide: clinical importance and proposed accurate investigative tool. *Human reproduction (Oxford, England)* 14, 1903-1907.
- Meirow, D., Lewis, H., Nugent, D., and Epstein, M. (1999b). Subclinical depletion of primordial follicular reserve in mice treated with cyclophosphamide: clinical importance and proposed accurate investigative tool. *Human Reproduction* 14, 1903-1907.
- Pascuali, N., Scotti, L., Di Pietro, M., Oubiña, G., Bas, D., May, M., Gómez Muñoz, A., Cuasnicú, P.S., Cohen, D.J., Tesone, M., *et al.* (2018). Ceramide-1-phosphate has protective properties against cyclophosphamide-induced ovarian damage in a mice model of premature ovarian failure. *Human reproduction (Oxford, England)* 33, 844-859.
- Qi, X., Yun, C., Sun, L., Xia, J., Wu, Q., Wang, Y., Wang, L., Zhang, Y., Liang, X., Wang, L., *et al.* (2019). Gut microbiota-bile acid-interleukin-22 axis orchestrates polycystic ovary syndrome. *Nature medicine* 25, 1225-1233.
- Tata, B., Mimouni, N.E.H., Barbotin, A.L., Malone, S.A., Loyens, A., Pigny, P., Dewailly, D., Catteau-Jonard, S., Sundström-Poromaa, I., Piltonen, T.T., *et al.* (2018). Elevated prenatal anti-Müllerian hormone reprograms the fetus and induces polycystic ovary syndrome in adulthood. *Nature medicine* 24, 834-846.
- Wang, S., Yao, H., Xu, Y., Hao, R., Zhang, W., Liu, H., Huang, Y., Guo, W., and Lu, B. (2020). Therapeutic potential of a TrkB agonistic antibody for Alzheimer's disease. *Theranostics* 10, 6854-6874.
- Yang, J.L., Lin, Y.T., Chuang, P.C., Bohr, V.A., and Mattson, M.P. (2014). BDNF and exercise enhance neuronal DNA repair by stimulating CREB-mediated production of apurinic/aprimidinic endonuclease 1. *Neuromolecular medicine* 16, 161-174.
- Yuan, X., Hu, T., Zhao, H., Huang, Y., Ye, R., Lin, J., Zhang, C., Zhang, H., Wei, G., Zhou, H., *et al.* (2016). Brown adipose tissue transplantation ameliorates polycystic ovary syndrome. *Proceedings of the National Academy of Sciences of the United States of America* 113, 2708-2713.

REVIEWER COMMENTS

Reviewer #3 (Remarks to the Author):

Thank you addressing my previous comments.

Questions about new data/text:

Introduction line 68-74. It should be clarified that the Kawamura et al PNAS 2013 paper cited reports the use of drug treatment (to stimulate Akt) to promote follicle activation/development as a treatment for POF. It does not report on cell death mechanisms of follicle depletion, as implied by the text as written.

Supplemental Figure 6a- while I appreciate the inclusion of human data (and this reviewer commends the effort taken), I am confused by this experiment. The legend indicates that the analysis is of BDNF relative intensity in follicles. How were follicles obtained analysed 61-73 year old women? At that age their ovaries would be devoid of follicles.

Supplemental Figure 6b. How were the follicle numbers within the tissues normalised? Wouldn't it be mostly stroma?

Outstanding issues:

Thank you addressing my initial comment about the follicle counting method. Can the authors please comment on the accuracy of the preantral and antral follicle counts in Figure 3g. If I understand the method correctly, follicles were counted in every 5th section, then this number multiplied by 5. This means, that on average, only 3 follicles for controls and 5 or so follicles for antibody treated mice were counted in the entire ovary for each follicle stage. Cell counting methodology (e.g. stereology) indicates that 100 objects need to be counted for accurate estimations and comparisons to be made. This reviewer appreciates that there are probably not 100 follicles present to count, nor was stereological technique used. But, with such low raw numbers counted, and a very small difference in the raw numbers between groups, I am not sure that there is a difference between groups. I am also not sure how one of the control mice could have fewer than 5 antral follicles using with method?

Figure 5. The authors have answered my previous question in relation to the rescue of growing follicles from atresia (although increased apoptosis granulosa cells 13 days after cyclophosphamide treated is quite surprising). However, it is still unclear how administration of Ab4B19 7 days after Cy treatment could rescue primordial follicles? The literature suggests that the depletion of primordial follicles is very rapid (whether it be by activation or apoptosis) i.e. based on current knowledge it seems likely that primordial follicle depletion would occur well before delivery of the antibody. If the authors are suggesting that Cy-induced primordial follicle depletion continues to occur after 7 days, then this should be demonstrated by showing fewer primordial follicles in Cy treated animals at 6 days vs 0 days (using the timeline in the experimental paradigm depicted in Figure 5a). Furthermore, Figure 5c is quite perplexing as it looks like there is no significant difference between the control and cy+Ab4B19 group, suggesting there is no primordial follicle depletion before antibody administration (i.e. in the first 7 days after cyclo). This finding suggests a full rescue, rather than a partial rescue. What was the statistical analysis applied to all the follicle counts- this info seems to be missing from the legend?

Figure 6 Heading-The Cyclo treated mice were not infertile. All mice in this study were fertile, in terms of all getting pregnant and delivering pups (Sup Table 6). Thus, I suggest this heading be changed to reflect that some fertility parameters were improved (i.e. superovulation and litter size), rather than "infertility reversed".

Figure 6H- The legend indicates that 21, 15 and 18 dams were present in each group but the dots on

the graphs and Sup Table 6 indicates there were 6 per group.

Response to reviews

Reviewer #3 (Remarks to the Author):

Thank you addressing my previous comments.

Questions about new data/text:

Introduction line 68-74. It should be clarified that the Kawamura et al PNAS 2013 paper cited reports the use of drug treatment (to stimulate Akt) to promote follicle activation/development as a treatment for POF. It does not report on cell death mechanisms of follicle depletion, as implied by the text as written.

Answer: We thank the reviewer for pointing out the inaccurate statement, and have now modified it to:

“Thus, these efforts are not very effective. One interesting study demonstrated that Akt-activating drugs could activate dormant primordial follicles and stimulate follicle growth. (Kawamura et al., 2013). However, this treatment involves surgical removal of a piece of ovary, fragmenting ovary into cubes for culture, followed by Akt stimulator treatment to activate primordial follicles. The activated donor ovarian cubes were then transplanted back to the recipient’s ovary, and subsequent retrieval of mature eggs for in vitro fertilization (IVF)”.

Supplemental Figure 6a- while I appreciate the inclusion of human data (and this reviewer commends the effort taken), I am confused by this experiment. The legend indicates that the analysis is of BDNF relative intensity in follicles. How were follicles obtained analysed 61-73 year old women? At that age their ovaries would be devoid of follicles.

Answer: It is generally agreed that the primordial follicle pool is almost exhausted after menopause. Thus, one could infer that there should be few follicles in the ovaries of women aged over 60. A previous report showed that the number of follicles decreased to less than 1000 at about 51, the average age when women enter menopause (Faddy et al., 1992), but the authors did not analyze the follicle numbers in those older than 51. It is unclear whether the remaining follicles were inactive, or could continue to develop. However, there are occasional reports that at age 60 or even 70, women could still get pregnant, suggesting that some of their follicles to certain extent might still be functional after amenorrhea. We performed careful examination and captured a few small follicles (diameter<100 µm) in women aged over 60. Despite of their poor morphology, they exhibited clear structural characteristics of primordial or primary follicles.

To determine whether BDNF expression in ovaries changes during aging process, we quantified the fluorescent intensity of BDNF immunostaining in follicle sections derived from women of different ages. Detailed tissue processing, staining and quantification procedures have now been provided in the Method section and in the

legend of Supplemental Figure 6. For each age group, we stained at least five sections. Careful examination identified at least 6 small follicles for women at ages 61, 63, and 64. For the 73-year old samples, we could find only four follicles in the 5 sections we examined.

Supplemental Figure 6b. How were the follicle numbers within the tissues normalised? Wouldn't it be mostly stroma?

Answer: The ovarian tissues were derived from the ovarian surgery of pre-menopausal women. Through dissecting microscope, we tried to remove fibrous tissues of the tunica albuginea and some adipose tissues. The dissected tissues with follicles were then fragmented, treated with vehicles, BDNF or Ab4B19. Since the ovarian pieces were mixed and divided into three groups, similar number of follicles were distributed in each group. While the ovarian tissues from surgery did contain a large amount of stroma, the strong activation of TrkB downstream signal Erk suggests there was a sufficient number of follicles in these tissues.

Outstanding issues:

Thank you addressing my initial comment about the follicle counting method. Can the authors please comment on the accuracy of the preantral and antral follicle counts in Figure 3g. If I understand the method correctly, follicles were counted in every 5th section, then this number multiplied by 5. This means, that on average, only 3 follicles for controls and 5 or so follicles for antibody treated mice were counted in the entire ovary for each follicle stage. Cell counting methodology (e.g. stereology) indicates that 100 objects need to be counted for accurate estimations and comparisons to be made. This reviewer appreciates that there are probably not 100 follicles present to count, nor was stereological technique used. But, with such low raw numbers counted, and a very small difference in the raw numbers between groups, I am not sure that there is a difference between groups. I am also not sure how one of the control mice could have fewer than 5 antral follicles using with method?

Answer: In "Methods" of initial submission and 1st revision, we have already indicated "For NA-POF model, follicle counting from all sections were performed." That is, all sections from an entire ovary were used, and we counted all follicles of different stages in an ovary, rather than counting only one fifth of them. We apologize that we might have not described the experimental method clearly, and now changed the description to make it clearer. It should be pointed out that, for female mice of 13 months old, the number of follicles is indeed very small. The follicles from each stage (from primordial to antral) mount only a few to dozens. Similar results have been reported before (Ding et al., 2018; Tamura et al., 2017).

Figure 5. The authors have answered my previous question in relation to the rescue of growing follicles from atresia (although increased apoptosis granulosa cells 13 days after cyclophosphamide treated is quite surprising). However, it is still unclear how

administration of Ab4B19 7 days after Cy treatment could rescue primordial follicles? The literature suggests that the depletion of primordial follicles is very rapid (whether it be by activation or apoptosis) i.e. based on current knowledge it seems likely that primordial follicle depletion would occur well before delivery of the antibody. If the authors are suggesting that Cy-induced primordial follicle depletion continues to occur after 7 days, then this should be demonstrated by showing fewer primordial follicles in Cy treated animals at 6 days vs 0 days (using the timeline in the experimental paradigm depicted in Figure 5a). Furthermore, Figure 5c is quite perplexing as it looks like there is no significant difference between the control and cy+Ab4B19 group, suggesting there is no primordial follicle depletion before antibody administration (i.e. in the first 7 days after cyclo). This finding suggests a full rescue, rather than a partial rescue. What was the statistical analysis applied to all the follicle counts- this info seems to be missing from the legend?

The reviewer's question includes the following major points, which we will address one by one:

1. The authors have answered my previous question in relation to the rescue of growing follicles from atresia (although increased apoptosis granulosa cells 13 days after cyclophosphamide treated is quite surprising).

Answer: As we responded previously (response to Reviewer#3, comment 12, Part 2-4), our results have shown that two weeks after Cy treatment, there were still significant apoptotic signals in the ovary. This is due possibly to follicular atresia, the secondary damage caused by the early death of granulosa cells. Similar findings (increased apoptotic signals two weeks after Cy treatment) have been reported by others (Pascuali et al., 2018). This point has been explained and discussed in the Discussion (page 15, manuscript with track change).

2. However, it is still unclear how administration of Ab4B19 7 days after Cy treatment could rescue primordial follicles? The literature suggests that the depletion of primordial follicles is very rapid (whether it be by activation or apoptosis) i.e. based on current knowledge it seems likely that primordial follicle depletion would occur well before delivery of the antibody. If the authors are suggesting that Cy-induced primordial follicle depletion continues to occur after 7 days, then this should be demonstrated by showing fewer primordial follicles in Cy treated animals at 6 days vs 0 days (using the timeline in the experimental paradigm depicted in Figure 5a).

Answer: The question "how administration of Ab4B19 7 days after Cy treatment could reduce primordial follicles?" has been also elaborated in our response to Reviewer#3, comment 12 (Part 2-4, especially 4), and now in the Discussion (page 15-16, manuscript with track change). In brief, Ab4B19 promotes the follicle development and protects developing follicles from apoptosis, thereby increasing AMH expression and inhibiting the depletion of primordial follicles. AMH level is critical for maintaining the inactive state of primordial follicles, which has been extensively studied (La Marca et al., 2009).

We agree that Cy-induced depletion of primordial follicles is very rapid. It has been shown that Cy treatment at 150 mg / kg leads to rapid apoptosis and PI3K-Akt

activation (1-3 days) (Kalich-Philosoph et al., 2013). However, there might be a secondary damage after this rapid apoptosis, as we had explained in the 1st revision. Both previous reports and our own data indicate that apoptotic signals last for at least 2 weeks. There are also significant reductions in various developing follicles and AMH levels, leading to a continuous loss of primordial follicles. Thus, the consumption of primordial follicles would always be higher than those in untreated ovaries until the AMH level returns to normal, and the duration may be much more than 13 (7+6) days. Our results that the delayed treatment of Ab4B19 (7 days after Cy treatment) can still partially rescue the damage also indirectly verify this secondary Cy-induced damage. Similar experiments in previous reports also showed a long-lasting depletion of primordial follicles induced by Cy treatment: 1) Depletion of primordial follicles induced by 50 mg/kg Cy lasted for at least 20 days in 18-week old mice (Fig. 4, (Kamarzaman et al., 2014)). 2) Depletion of primordial follicles caused by 150 mg/kg Cy treatment on day-21 was much more obvious than that on day-7 (~45% .vs. 20%) (Fig. 2a and Fig. 2d, (Roness et al., 2019)).

3. Furthermore, Figure 5c is quite perplexing as it looks like there is no significant difference between the control and cy+Ab4B19 group, suggesting there is no primordial follicle depletion before antibody administration (i.e in the first 7 days after cyclo). This finding suggests a full rescue, rather than a partial rescue.

Answer: There is indeed a significant difference between the control and Cy+Ab4B19 groups in Figure 5c. However, we did not mark it because the comparison is less relevant to the main theme of the paper. The primordial follicles in Cy+Ab4B19 group (461.9 ± 29.80) is 20.4% lower than that in control (580.0 ± 42.58), and $P = 0.0393$, showing a continuous depletion of primordial follicles in the first 7 days of Cy treatment. Therefore, our results revealed a partial rescue, not a full rescue, of primordial follicles by Ab4B19.

4. What was the statistical analysis applied to all the follicle counts- this info seems to be missing from the legend?

Answer: We agree that we did not indicate the statistical method clearly, and have now added the following in Fig. 5:

“Unless stated otherwise, statistical analyses for comparison among 3 or more groups in this and all the following figures were carried out using one-way ANOVA, followed by Dunnett's multiple comparisons test”.

Figure 6 Heading-The Cyclo treated mice were not infertile. All mice in this study were fertile, in terms of all getting pregnant and delivering pups (Sup Table 6). Thus, I suggest this heading be changed to reflect that some fertility parameters were improved (i.e. superovulation and litter size), rather than “infertility reversed”.

Answer: We thank the reviewer for the suggestion and have changed heading to:

“Improvement in fertility by Ab4B19 in Cy-POF model.” .

Figure 6H- The legend indicates that 21, 15 and 18 dams were present in each group but the dots on the graphs and Sup Table 6 indicates there were 6 per group.

Answer: The reviewer was right that in each condition (Ctrl, Vehicle and Ab4B19-treated), we used 6 adult female mice per group. The 6 female mothers were mated 3-4 times with male mice in their life time (mating and breeding were terminated when the female mothers were approximately 9 months old), and the litter size (how many babies per birth) in the 1st and 2nd rounds of mating was presented in in Fig. 6h and 6i, respectively. On average, there were approximately 10 babies/mother for control, 6 babies/mother for Cy, and 9 babies/mother for Cy-Ab4B19 groups in both 1st and 2nd rounds, indicating clearly that the TrkB antibody treatment improved fertility. Moreover, the six female mice in the Ctrl group had given altogether 21 times of birth, averaging 3.5 births/mouse before termination (in 9 months). The Vehicle and Ab4B19-treated groups actually had completed 15 times of birth ($15/6=2.5$ births/mouse) and 18 times of birth ($18/6=3$ births/mouse) before termination, respectively. Thus, Ab4B19 not only increased number of pups/birth but also number of births in their adult life (9 months). We apologize for the inaccurate expression, and have now corrected these in both the Results and the legend of Fig. 6h, 6i.

Reference:

Ding, C., Zou, Q., Wang, F., Wu, H., Wang, W., Li, H., and Huang, B. (2018). HGF and BFGF Secretion by Human Adipose-Derived Stem Cells Improves Ovarian Function During Natural Aging via Activation of the SIRT1/FOXO1 Signaling Pathway. *Cellular physiology and biochemistry : international journal of experimental cellular physiology, biochemistry, and pharmacology* 45, 1316-1332.

Faddy, M.J., Gosden, R.G., Gougeon, A., Richardson, S.J., and Nelson, J.F. (1992). Accelerated disappearance of ovarian follicles in mid-life: implications for forecasting menopause. *Human reproduction (Oxford, England)* 7, 1342-1346.

Kalich-Philosoph, L., Roness, H., Carmely, A., Fishel-Bartal, M., Ligumsky, H., Paglin, S., Wolf, I., Kanety, H., Sredni, B., and Meirou, D. (2013). Cyclophosphamide triggers follicle activation and "burnout"; AS101 prevents follicle loss and preserves fertility. *Science translational medicine* 5, 185ra162.

Kamarzaman, S., Shaban, M., and Rahman, S.A. (2014). The prophylactic effect of *Nigella sativa* against cyclophosphamide in the ovarian follicles of matured adult mice: a preliminary study. *JAPS, Journal of Animal and Plant Sciences* 24, 81-88.

Kawamura, K., Cheng, Y., Suzuki, N., Deguchi, M., Sato, Y., Takae, S., Ho, C.H., Kawamura, N., Tamura, M., Hashimoto, S., *et al.* (2013). Hippo signaling disruption and Akt stimulation of ovarian follicles for infertility treatment. *Proceedings of the National Academy of Sciences of the United States of America* 110, 17474-17479.

La Marca, A., Broekmans, F.J., Volpe, A., Fauser, B.C., and Macklon, N.S. (2009). Anti-Mullerian hormone (AMH): what do we still need to know? *Human reproduction* (Oxford, England) *24*, 2264-2275.

Pascuali, N., Scotti, L., Di Pietro, M., Oubiña, G., Bas, D., May, M., Gómez Muñoz, A., Cuasnicú, P.S., Cohen, D.J., Tesone, M., *et al.* (2018). Ceramide-1-phosphate has protective properties against cyclophosphamide-induced ovarian damage in a mice model of premature ovarian failure. *Human reproduction* (Oxford, England) *33*, 844-859.

Roness, H., Spector, I., Leichtmann-Bardoogo, Y., Savino, A.M., Dereh-Haim, S., and Meirou, D. (2019). Pharmacological administration of recombinant human AMH rescues ovarian reserve and preserves fertility in a mouse model of chemotherapy, without interfering with anti-tumoural effects. *Journal of Assisted Reproduction and Genetics* *36*, 1793-1803.

Tamura, H., Kawamoto, M., Sato, S., Tamura, I., Maekawa, R., Taketani, T., Aasada, H., Takaki, E., Nakai, A., Reiter, R.J., *et al.* (2017). Long-term melatonin treatment delays ovarian aging. *Journal of pineal research* *62*.

REVIEWER COMMENTS

Reviewer #3 (Remarks to the Author):

Unfortunately, I am not convinced by available data or the explanation for how the antibody could prevent primordial follicle depletion (due to "secondary damage") when given so long after cyclo treatment. To demonstrate that Cyclophosphamide-induced primordial follicle depletion continues to occur after 7 days, follicle numbers should be determined at Day 0 (the first time point at which the antibody is administered) for each group, and then compared with the 6 day time point (using the timeline in the experimental paradigm depicted in Figure 5a). A decline in primordial follicle numbers should be observed in cyclo treated animals during this time and this decline should be prevented by the antibody. Relating to this issue, in Line 477 it is stated that cyclophosphamide was found to cause a "continuous loss of primordial follicles". Continuous loss of primordial follicles was not demonstrated, and in the absence of the requested additional data, the sentences that follow are too speculative.

Figure 7 F. Based on morphology alone, it is unclear if the stained structures are in fact follicles, particularly in the aged group where follicles would be extremely rare (if present at all). Co-labelling with an established marker, like MVH, is required to confirm the identity of these structures.

Reviewer #3 (Remarks to the Author):

Unfortunately, I am not convinced by available data or the explanation for how the antibody could prevent primordial follicle depletion (due to "secondary damage") when given so long after cyclo treatment. To demonstrate that Cyclophosphamide-induced primordial follicle depletion continues to occur after 7 days, follicle numbers should be determined at Day 0 (the first time point at which the antibody is administered) for each group, and then compared with the 6 day time point (using the timeline in the experimental paradigm depicted in Figure 5a). A decline in primordial follicle numbers should be observed in cyclo treated animals during this time and this decline should be prevented by the antibody. Relating to this issue, in Line 477 it is stated that cyclophosphamide was found to cause a "continuous loss of primordial follicles". Continuous loss of primordial follicles was not demonstrated, and in the absence of the requested additional data, the sentences that follow are too speculative.

Answer: We appreciate that Reviewer#3 wanted us to provide "additional data" demonstrating that "Cyclophosphamide-induced primordial follicle depletion continues to occur after 7 days", instead of citing two previous publications showing the similar results (Kamarzaman et al., 2014); (Rones et al., 2019). We have now performed this experiment as requested and the data indicated the Cy-induced continued depletion of primordial follicles after 7 days (Supplementary Fig. 4a). We have summarized the following three points to further clarify this issue:

1. Our data showed that the remaining primordial follicles in Cy-treated mice was approximately 63% of that in untreated mice at 7-day time point, whereas only about 45% at 13-day (7+6) time point (Supplementary Fig. 4a). Thus, a further decline in primordial follicle numbers was observed in Cy-treated mice during the 6-day period, and this decline was attenuated by the TrkB antibody (Fig. 5c, left).
2. Previous publications also reported similar results: 1) Depletion of primordial follicles induced by 50 mg/kg Cy lasted for at least 20 days in 18-week old mice (Fig. 4, in (Kamarzaman et al., 2014)). 2) Depletion of primordial follicles caused by 150 mg/kg Cy treatment was much more obvious on day-21 than that on day-7 (~45% .vs. 20%) (see Fig. 2a and Fig. 2d in (Rones et al., 2019)).
3. The phenotype of "continuous Cy-induced primordial follicle depletion" is also reasonable from a mechanistic point of view. It has been well demonstrated that a rapid (1-3 days) Cy-induced primordial follicle depletion is mediated by an activation of PI3K-Akt signaling pathway (direct effect) in primordial follicles. However, Cy-induced depletion is not only due to the direct effect. Other factors may also have some indirect effects on primordial follicle depletion. For example, AMH secreted by GCs could inhibit the activation and therefore maintain the pool of primordial follicles. Cy-treatment induce a significant reduction in AMH levels, leading to a continuous loss of primordial follicles. Ab4B19 was able to reverse the decrease in AMH (Fig. 6b). In addition, it has been proposed that neighboring primordial follicles could elicit inhibitory effects on themselves (Da Silva-Buttkus et al., 2009). This may also contribute to a continuous decline of primordial follicles after 7 days of Cy-treatment. Thus, a rapid, PI3K-Akt-mediated depletion of primordial follicles may be followed by

a long-lasting secondary decline of primordial follicle number, and this effect could be reversed by the TrkB agonistic antibody. These direct and indirect effects have been suggested by a recent review article (Spears et al., 2019).

Figure 7 F. Based on morphology alone, it is unclear if the stained structures are in fact follicles, particularly in the aged group where follicles would be extremely rare (if present at all). Co-labelling with an established marker, like MVH, is required to confirm the identity of these structures.

Answer: We thank Reviewer#3 for the suggestion. Although the follicles were traditionally identified by the structures, we have done the co-immunostaining experiment as requested. We used an established follicle marker DDX4/MVH. The new data have been shown in Fig. 5e, f and Supplementary Fig. 6b. Please note, the 73-year old sample was not included in this revision because very few DDX4-positive follicles could be detected. Further, we also showed that three DDX4-positive follicles could be found in one section from a 61-year-old female (Supplementary Fig. 6b). Therefore, the remaining follicles in aged females might be more than estimated, despite of their poor morphology.

Reference:

- Da Silva-Buttkus, P., Marcelli, G., Franks, S., Stark, J., and Hardy, K. (2009). Inferring biological mechanisms from spatial analysis: Prediction of a local inhibitor in the ovary. *106*, 456-461.
- Kamarzaman, S., Shaban, M., and Rahman, S.A. (2014). The prophylactic effect of *Nigella sativa* against cyclophosphamide in the ovarian follicles of matured adult mice: a preliminary study. *JAPS, Journal of Animal and Plant Sciences* 24, 81-88.
- Roness, H., Spector, I., Leichtmann-Bardoogo, Y., Savino, A.M., Dereh-Haim, S., and Meirow, D. (2019). Pharmacological administration of recombinant human AMH rescues ovarian reserve and preserves fertility in a mouse model of chemotherapy, without interfering with anti-tumoural effects. *Journal of Assisted Reproduction and Genetics* 36, 1793-1803.
- Spears, N., Lopes, F., Stefansdottir, A., Rossi, V., De Felici, M., Anderson, R.A., and Klinger, F.G. (2019). Ovarian damage from chemotherapy and current approaches to its protection. *Human Reproduction Update* 25, 673-693.

REVIEWER COMMENTS

Reviewer #3 (Remarks to the Author):

Thank you for addressing my concerns by doing additional experiments.

Reviewer #3 (Remarks to the Author):

Thank you for addressing my concerns by doing additional experiments.

Answer: We thank the reviewer for his/her approval of our revision.